# Iterative Reachability Estimation for Safe Reinforcement Learning

**Milan Ganai**
UC San Diego
mganai@ucsd.edu

**Zheng Gong**
UC San Diego
zhgong@ucsd.edu

**Chenning Yu**
UC San Diego
chy010@ucsd.edu

**Sylvia Herbert**
UC San Diego
sherbert@ucsd.edu

**Sicun Gao**
UC San Diego
sicung@ucsd.edu

## Abstract

Ensuring safety is important for the practical deployment of reinforcement learning (RL). Various challenges must be addressed, such as handling stochasticity in the environments, providing rigorous guarantees of persistent state-wise safety satisfaction, and avoiding overly conservative behaviors that sacrifice performance. We propose a new framework, Reachability Estimation for Safe Policy Optimization (RESPO), for safety-constrained RL in general stochastic settings. In the feasible set where there exist violation-free policies, we optimize for rewards while maintaining persistent safety. Outside this feasible set, our optimization produces the safest behavior by guaranteeing entrance into the feasible set whenever possible with the least cumulative discounted violations. We introduce a class of algorithms using our novel reachability estimation function to optimize in our proposed framework and in similar frameworks such as those concurrently handling multiple hard and soft constraints. We theoretically establish that our algorithms almost surely converge to locally optimal policies of our safe optimization framework. We evaluate the proposed methods on a diverse suite of safe RL environments from Safety Gym, PyBullet, and MuJoCo, and show the benefits in improving both reward performance and safety compared with state-of-the-art baselines.

## 1 Introduction

Safety-Constrained Reinforcement Learning is important for a multitude of real-world safety-critical applications [1, 2]. These situations require guaranteeing persistent safety and achieving high performance while handling environment uncertainty. Traditionally, methods have been proposed to obtain optimal control by solving within the Constrained Markov Decision Process (CMDP) framework, which constrains expected cumulative violation of the constraints along trajectories to be under some threshold. Trust-region approaches [3, 4, 5] follow from the Constrained Policy Optimization algorithm [6] which tries to guarantee monotonic performance improvement while satisfying the cumulative constraint requirement. However, these approaches are generally very computationally expensive, and their first or second order Taylor approximations may have repercussions on performance [2]. Several primal-dual approaches [7, 8, 9] have been proposed with better computational efficiency and use multi-timescale frameworks [10] for asymptotic convergence to a local optimum. Nonetheless, these approaches inherit CMDP's lack of crucial rigorous safety guarantees that ensure persistent safety. Particularly, maintaining a cost budget for hard constraints like collision avoidance is insufficient. Rather, it's crucial to prioritize obtaining safe, minimal-violation policies when possible.

37th Conference on Neural Information Processing Systems (NeurIPS 2023).

There have been a variety of proposed control theoretic functions that provide the needed guarantees on persistent safety for such hard constraints, including Control Barrier Functions (CBFs) [11] and Hamilton-Jacobi (HJ) reachability analysis [12, 13]. CBFs are energy-based certification functions, whose zero super-level set is control invariant [14, 15]. The drawbacks are the conservativeness of the feasible set and the lack of a systematic way to find these functions for general nonlinear systems. Under certain assumptions, approximate energy-based certificates can be learned with sufficient samples [16, 17, 18, 19, 20, 21]. HJ reachability analysis focuses on finding the set of initial states such that there exists some control to avoid and/or reach a certain region. It computes a value function, whose zero sub-level set is the largest feasible set, together with the optimal control. The drawback is the well known 'curse of dimensionality,' as HJ reachability requires solving a corresponding Hamilton-Jacobi-Bellman Partial Differential Equation (HJBPDE) on a discrete grid recursively. Originally, both approaches assume the system dynamics and environment are known, and inputs are bounded by some polytopic constraints, while recent works have applied them to model-free settings [22, 23, 24]. For both CBF and HJ reachability, controllers that optimize performance and guarantee safety can be synthesized in different ways online [25]. There also exist hard-constraint approaches like [26] have theoretical safety guarantees without relying on control theory; however, they require learning or having access to the transition-function dynamics.

Consequently, it can be beneficial to combine CMDP-based approaches to find optimal control and the advantages of control-theoretic approaches to enable safety. The recent framework of Reachability Constrained Reinforcement Learning (RCRL) [27] proposed an approach that guarantees persistent safety and optimal performance when the agent is in the feasible set. However, RCRL is *limited to learning deterministic policies in deterministic environments*. In the general RL setting, stochasticity incurs a variety of challenges including how to define membership in the feasible set since it is no longer a binary question but rather a probabilistic one. Another issue is that for states outside optimal feasible set, RCRL optimization *cannot guarantee entrance into the feasible set when possible*. This may have undesirable consequences including indefinitely violating constraints with no recovery.

In light of the above challenges, we propose a new framework and class of algorithms called Reachability Estimation for Safe Policy Optimization (**RESPO**). Our main contributions are:

• We introduce reachability estimation methods for general stochastic policies that predict the likelihood of constraint violations in the future. Using this estimation, a policy can be optimized such that **(i)** when in the feasible set, it maintains persistent safety and optimizes for reward performance, and **(ii)** when outside the feasible set, it produces the safest behavior and enters the feasible set if possible. We also demonstrate how our optimization can incorporate multiple hard and soft constraints while even prioritizing the hard constraints.

• We provide a novel class of actor-critic algorithms based on learning our reachability estimation function, and prove our algorithm converges to a locally optimal policy of our proposed optimization.

• We perform comprehensive experiments showing that our approach achieves high performance in complex, high-dimensional Safety Gym, PyBullet-based, and MuJoCo-based environments compared to other state-of-the-art safety-constrained approaches while achieving small or even $0$ violations. We also show our algorithm's performance in environments with multiple hard and soft constraints.

## 2 Related Work

### 2.1 Constrained Reinforcement Learning

Safety-constrained reinforcement learning has been addressed through various proposed learning mechanisms solving within an optimization framework. CMDP [28, 29] is one framework that augments MDPs with the cost function: the goal is to maximize expected reward returns while constraining cost returns below a manually chosen threshold. Trust-region methods [3, 4, 5, 6] and primal-dual based approaches [7, 9, 30, 31] are two classes of CMDP-based approaches. Trust-region approaches generally follow from Constrained Policy Optimization (CPO) [6], which approximates the constraint optimization problem with surrogate functions for the reward objective and safety constraints, and then performs projection steps on the policy parameters with backtracking line search. Penalized Proximal Policy Optimization [32] improves upon past trust-region approaches via the clipping mechanism, similar to Proximal Policy Optimization's [33] improvement over Trust Region Policy Optimization [34]. Constraint-Rectified Policy Optimization [35] takes steps toward improving reward performance if constraints are currently satisfied else takes steps to minimize

constraint violations. PPO Lagrangian [30] is a primal-dual method combining Proximal Policy Optimization (PPO) [33] with lagrangian relaxation of the safety constraints and has relatively low complexity but outperforms CPO in constraint satisfaction. [7] uses a penalized reward function like PPO Lagrangian, and demonstrates convergence to an optimal policy through the multi-timescale framework, introduced in [10]. [36] proposes mapping the value function constraining problem as constraining densities of state visitation. Overall, while these algorithms provide benefits through various learning mechanisms, they inherit the problems of the CMDP framework. Specifically, they do not provide rigorous guarantees of persistent safety and so are generally unsuitable for state-wise constraint optimization problems.

## 2.2 Hamilton-Jacobi Reachability Analysis

HJ reachability analysis is a rigorous approach that verifies safety or reachability for dynamical systems. Given any deterministic nonlinear system and target set, it computes the viscosity solution for the HJBPDE, whose zero sub-level set is the backward reachable set (BRS), meaning there exists some control such that the states starting from this set will enter the target set in future time (or stay away from the target set for all future time). However, the curse of dimensionality and the assumption of knowing the dynamics and environment in advance restrict its application, especially in the model-free RL setting where there is no access to the entire observation space and dynamics at any given time. Decomposition [37], warm-starting [38], sampling-based reachability [39], and Deepreach [40] have been proposed to solve the curse of dimensionality. [23, 24] proposed methods to bridge HJ analysis with RL by modifying the HJBPDE. The work of [27] takes framework into a deterministic hard-constraint setting. There are additionally model-based HJ reachability approach for reinforcement learning-based controls. [41] is a model-based extension of [27]. Approaches like [42, 43, 44] use traditional HJ reachability for RL control while learning or assuming access to the system dynamics model. In stochastic systems, finding the probability of reaching a target while avoiding certain obstacles are key problems. [45, 46] constructed the theoretic framework based on dynamic programming and consider finite and infinite time reach-avoid problems. [47, 48, 49] propose computing the stochastic reach-avoid set together with the probability in a tractable manner to address the curse of dimensionality.

## 3 Preliminaries

### 3.1 Markov Decision Processes

Markov Decision Processes (MDP) are defined as $\mathcal{M} := \langle \mathcal{S}, \mathcal{A}, P, r, h, \gamma \rangle$. $\mathcal{S}$ and $\mathcal{A}$ are the state and action spaces respectively. $P : \mathcal{S} \times \mathcal{A} \times \mathcal{S} \mapsto [0, 1]$ is the transition function capturing the environment dynamics. $r : \mathcal{S} \times \mathcal{A} \mapsto \mathbb{R}$ is the reward function associated with each state-action pair, $h : \mathcal{S} \mapsto \mathbb{R}_0^+$ is the safety loss function that maps a state to a non-negative real value, which is called the constraint value, or simply cost. $H_{\min}$ is the minimum *non-zero* value of function $h$; $H_{\max}$ is upper bound on function $h$. $\gamma$ is a discount factor in the range $(0, 1)$. $\mathcal{S}_I$ is initial state set, $d_0$ is initial state distribution, and $\pi(a|s)$ is a stochastic policy that is parameterized by the state and returns an action distribution from which an action can be sampled and affects the environment defined by the MDP. In unconstrained RL, the goal is to learn an optimal policy $\pi^*$ maximizing expected discounted sum of rewards, i.e. $\pi^* = \arg\max_\pi \mathbb{E}_{s \sim d_0} V^\pi(s)$, where $V^\pi(s) := \mathbb{E}_{\tau \sim \pi, P(s)}[\sum_{s_t \in \tau} \gamma^t r(s_t, a_t)]$. Note: $\tau \sim \pi, P(s)$ indicates sampling trajectory $\tau$ for horizon $T$ starting from state $s$ using policy $\pi$ in MDP with transition model $P$, and $s_t \in \tau$ is the $t^{th}$ state in trajectory $\tau$.

### 3.2 Constrained Markov Decision Process

CMDP attempts to optimize the reward returns $V^\pi(s)$ under the constraint that the cost return is below some manually chosen threshold $\chi$. Specifically:

$$\max_\pi \mathbb{E}_{s \sim d_0} [V^\pi(s)], \text{ subject to } \mathbb{E}_{s \sim d_0} [V_c^\pi(s)] \leq \chi, \qquad \text{(CMDP)}$$

where cost return function $V_c^\pi(s)$ is often defined as $V_c^\pi(s) := \mathbb{E}_{\tau \sim \pi, P(s)}[\sum_{s_t \in \tau} \gamma^t h(s_t)]$.

While many approaches have been proposed to solve within this framework, CMDPs have several difficulties: 1. cost threshold $\chi$ often requires much tuning while using prior knowledge of the environment; and 2. CMDP often permits some positive average cost which is incompatible with state-wise hard constraint problems, since $\chi$ is usually chosen to be above 0.

# 4 Stochastic Hamilton-Jacobi Reachability for Reinforcement Learning

Classic HJ reachability considers finding the largest feasible set for deterministic environments. In this section, we apply a similar definition in [45, 46] and define the stochastic reachability problem.

## 4.1 Persistent Safety and HJ Reachability for Stochastic Systems

The instantaneous safety can be characterized by the safe set $\mathcal{S}_s$, which is the zero level set of the safety loss function $h : \mathcal{S} \mapsto \mathbb{R}_0^+$. The unsafe (i.e. violation) set $\mathcal{S}_v$ is the complement of the safe set.

**Definition 1.** Safe set and unsafe set: $\mathcal{S}_s := \{s \in \mathcal{S} : h(s) = 0\}, \mathcal{S}_v := \{s \in \mathcal{S} : h(s) > 0\}$.

We will write $\mathbb{1}_{s \in \mathcal{S}_v}$ as the *instantaneous violation indicator function*, which is 1 if the current state is in the violation set and 0 otherwise. Note that the safety loss function $h$ is different from the instantaneous violation indicator function since $h$ captures the magnitude of the violation at the state.

It is insufficient to only consider instantaneous safety. When the environment and policy are both deterministic, we easily have a unique trajectory for starting from each state (i.e. the future state is uniquely determined) under Lipschitz environment dynamics. In classic HJ reachability literature [13], for a deterministic MDP's transition model $P_d$ and deterministic policy $\pi_d$, the set of states that guarantees persistent safety is captured by the zero sub-level set of the following value function:

**Definition 2.** Reachability value function $V_h^\pi : \mathcal{S} \mapsto \mathbb{R}_0^+$ is: $V_h^\pi(s) := \max_{s_t \in \tau \sim \pi_d, P_d(s)} h(s_t)$.

However, when there's a stochastic environment with transition model $P(\cdot|s, a)$ and policy $\pi(\cdot|s)$, the future states are not uniquely determined. This means for a given initial state and policy, there may exist many possible trajectories starting from this state. In this case, instead of defining a binary function that only indicates the existence of constraint violations, we define the reachability estimation function (REF), which captures the probability of constraint violation:

**Definition 3.** The reachability estimation function (REF) $\phi^\pi : \mathcal{S} \mapsto [0, 1]$ is defined as:

$$\phi^\pi(s) := \mathop{\mathbb{E}}_{\tau \sim \pi, P(s)} \max_{s_t \in \tau} \mathbb{1}_{(s_t | s_0 = s, \pi) \in \mathcal{S}_v}.$$

In a specific trajectory $\tau$, the value $\max_{s_t \in \tau} \mathbb{1}_{(s_t | s_0 = s, \pi) \in \mathcal{S}_v}$ will be 1 if there exist constraint violations and 0 if there exists no violation, which is binary. Taking expectation over this binary value for all the trajectories, we get the desired probability. We define optimal REF based on an optimally safe policy $\pi^* = \arg\min_\pi V_c^\pi(s)$ (note that this policy may not be unique).

**Definition 4.** The optimal reachability estimation function $\phi^* : \mathcal{S} \mapsto [0, 1]$ is: $\phi^*(s) := \phi^{\pi^*}(s)$.

Interestingly, we can utilize the fact the instantaneous violation indicator function produces binary values to learn the REF function in a bellman recursive form. The following will be used later:

**Theorem 1.** The REF can be reduced to the following recursive Bellman formulation:

$$\phi^\pi(s) = \max\{\mathbb{1}_{s \in \mathcal{S}_v}, \mathop{\mathbb{E}}_{s' \sim \pi, P(s)} \phi^\pi(s')\},$$

where $s' \sim \pi, P(s)$ is a sample of the immediate successive state (i.e., $s' \sim P(\cdot|s, a \sim \pi(\cdot|s))$) and the expectation is taken over all possible successive states. The proof can be found in the appendix.

**Definition 5.** The feasible set of a policy $\pi$ based on $\phi^\pi(s)$ is defined as: $\mathcal{S}_f^\pi := \{s \in \mathcal{S} : \phi^\pi(s) = 0\}$.

Note, the feasible set for a specific policy is the set of states starting *from* which no violation is reached, and the safe set is the set of states *at* which there is no violation. We will use the phrase likelihood of being feasible to mean the likelihood of not reaching a violation, i.e. $1 - \phi^\pi(s)$.

## 4.2 Comparison with RCRL

The RCRL approach [27] uses reachability to optimize and maintain persistent safety in the feasible set. Note, in below formulation, $\mathcal{S}_f$ is the optimal feasible set, i.e. that of a policy $\arg\min_\pi V_h^\pi(s)$. The RCRL formulation is:

$$\max_\pi \mathop{\mathbb{E}}_{s \sim d_0} [V^\pi(s) \cdot \mathbb{1}_{s \in \mathcal{S}_f} - V_h^\pi(s) \cdot \mathbb{1}_{s \notin \mathcal{S}_f}], \text{ subject to } V_h^\pi(s) \le 0, \forall s \in \mathcal{S}_I \cap \mathcal{S}_f. \qquad \text{(RCRL)}$$

The equation RCRL considers two different optimizations. When in the optimal feasible set, the optimization produces a persistently safe policy maximizing rewards. When outside this set, the

optimization produces a control minimizing the maximum future violation, i.e. $\arg\min_\pi V_h^\pi(s)$. *However, this does not ensure (re)entrance into the feasible set even if such a control exists.*

RCRL performs constraint optimization on $V_h^\pi$ with a neural network (NN) lagrange multiplier with state input [9]. When learning to optimize a Lagrangian dual function, the NN lagrange multiplier should converge to small values for states in the optimal feasible set and converge to large values for other states. Nonetheless, learning $V_h$ provides a weak signal during training: if there is an improvement in safety along the trajectory not affecting the maximum violation, $V_h^\pi$ remains the same for all states before the maximum violation in the trajectory. These improvements in costs can be crucial in guiding the optimization toward a safer policy. And optimizing with $V_h(s)$ can result in accumulating an unlimited number of violations smaller than the maximum violation. Also, a major issue with this approach is that *it's limited to deterministic MDPs and policies* because its reachability value function in the Bellman formulation does not directly apply to the stochastic setting. However, in general *stochastic* settings, estimating feasibility cannot be binary since for a large portion of the state space, even under the optimal policy, the agent may enter the unsafe set with a non-zero probability, rendering such definition too conservative and impractical.

# 5 Iterative Reachability Estimation for Safe Reinforcement Learning

In this paper, we formulate a general optimization framework for safety-constrained RL and propose a new algorithm to solve our constraint optimization by using our novel reachability estimation function. We present the deterministic case in Section 5.1 and build our way to the stochastic case in Section 5.2. We present our novel algorithm to solve these optimizations, involving our new reachability estimation function, in Section 5.3. We introduce convergence analysis in Section 5.4.

## 5.1 Iterative Reachability Estimation for Deterministic Settings

All state transitions and policies happen with likelihood $0$ or $1$ for the deterministic environment. Therefore, the probability of constraint violation for policy $\pi$ from state $s$, i.e., $\phi^\pi(s)$, is in the set $\{0, 1\}$. According to Definition 4, if there exists some policy $\pi$ such that $\phi^\pi(s) = 0$, we have $\phi^*(s) = 0$. Otherwise, $\phi^*(s) = 1$. Notice that this captures definitive membership in the optimal feasible set $\phi^*(s) = \mathbb{1}_{s \in S_f^{\pi_s}}$, which is the feasible set of some safest policy $\pi_s = \arg\min_\pi V_c^\pi(s)$. Now, we divide our optimization in two parts: the infeasible part and the feasible part.

For the infeasible part, we want the agent to incur the least cumulative damage (discounted sum of costs) and, if possible, (re)enter the feasible set. Different from previous Reachability-based RL optimizations, by using the discounted sum of costs $V_c^\pi(s)$ we consider both magnitude and frequency of violations, thereby improving learning signal. The infeasible portion takes the form:

$$\max_\pi \mathbb{E}_{s \sim d_0} \left[-V_c^\pi(s)\right]. \tag{1}$$

For the feasible part, we want the policy to ensure the agent stays in the feasible set and maximize reward returns. This produces a constraint optimization where the cost value function is constrained:

$$\max_\pi \mathbb{E}_{s \sim d_0} \left[V^\pi(s)\right], \text{ subject to } V_c^\pi(s) = 0, \forall s \in \mathcal{S}_I. \tag{2}$$

The following propositions justify using $V_c^\pi$ as the constraint. Both proofs are in the appendix.

**Proposition 1.** The cost value function $V_c^\pi(s)$ is zero for state $s$ if and only if the persistent safety is guaranteed for that state under the policy $\pi$.

We define here $\mathcal{S}_f := \mathcal{S}_f^{\pi_s}$, the feasibilty set of some safest policy. Now, the above two optimizations can be unified with the use of the feasibility function $\phi^*(s)$:

$$\max_\pi \mathbb{E}_{s \sim d_0} \left[V^\pi(s) \cdot (1 - \phi^*(s)) - V_c^\pi(s) \cdot \phi^*(s)\right], \text{ subject to } V_c^\pi(s) = 0, \forall s \in \mathcal{S}_I \cap \mathcal{S}_f. \tag{3}$$

Unlike other reachability based optimizations like RCRL, one particular advantage in Equation 3 is, with some assumptions, the guaranteed entrance back into feasible set with minimum cumulative discounted violations whenever a possible control exists. More formally, assuming infinite horizon:

**Proposition 2.** If $\exists \pi$ that produces trajectory $\tau = \{(s_i), i \in \mathbb{N}, s_1 = s\}$ in deterministic MDP $\mathcal{M}$ starting from state $s$, and $\exists m \in \mathbb{N}, m < \infty$ such that $s_m \in S_f^\pi$, then $\exists \epsilon > 0$ where if discount factor $\gamma \in (1 - \epsilon, 1)$, then the optimal policy $\pi^*$ of Equation 3 will produce a trajectory $\tau' = \{(s_j'), j \in \mathbb{N}, s_1' = s\}$, such that $\exists n \in \mathbb{N}, n < \infty, s_n' \in S_f^{\pi^*}$ and $V_c^{\pi^*}(s) = \min_{\pi'} V_c^{\pi'}(s)$.

## 5.2 Iterative Reachability Estimation for Stochastic Settings

In stochastic environments, for each state, there is some likelihood of entering into the unsafe states under any policy. Thus, we adopt the probabilistic reachability Definitions 3 and 4. Rather than using the binary indicator in the optimal feasible set to demarcate the feasibility and infeasibility optimization scenarios, we use the likelihood of infeasibility of the safest policy. In particular, for any state $s$, the optimal likelihood that the policy will enter the infeasible set is $\phi^*(s)$ from Definition 4.

We again divide the full optimization problem in stochastic settings into infeasible and feasible ones similar to Equations 1 and 2. However, we consider the infeasible formulation with likelihood the current state is in a safest policy's infeasible state, or $\phi^*(s)$. Similarly, we account for the feasible optimization formulation with likelihood the current state is in a safest policy's feasible set, $1 - \phi^*(s)$. The complete Reachability Estimation for Safe Policy Optimization (RESPO) can be rewritten as:

$$\max_{\pi} \mathbb{E}_{s \sim d_0} [V^\pi(s) \cdot (1 - \phi^*(s)) - V_c^\pi(s) \cdot \phi^*(s)], \text{ s.t., } V_c^\pi(s) = 0, \text{ w.p. } 1 - \phi^*(s), \forall s \in S_I.$$

$$\text{(RESPO)}$$

In sum, the RESPO framework provides several benefits when compared with other constrained Reinforcement Learning and reachability-based approaches. Notably, 1) it maintains persistent safety when in the feasible set unlike CMDP-based approaches, 2) compared with other reachability-based approaches, RESPO considers performance optimization in addition to maintaining safety, 3) it maintains the behavior of a safest policy in the infeasible set and even reenters the feasible set when possible, 4) RESPO employs rigorously defined reachability definitions even in stochastic settings.

## 5.3 Overall Algorithm

We describe our algorithms by breaking down the novel components. Our algorithm predicts reachability membership to guide the training toward optimizing the right portion of the optimization equation (i.e., feasibility case or infeasibility case). Furthermore, it exclusively uses the discounted sum of costs as the safety value function – we can avoid having to learn the reachability value function while having the benefit of exploiting the improved signal in the cost value function.

**Optimization in infeasible set versus feasible set.** If the agent is in the infeasible set, this is the simplest case. We want to find the optimal policy that maximizes $-V_c^\pi(s)$. This would be the only term that needs to be considered in optimization.

On the other hand, if the agent is in the feasible set, we must solve the constraint optimization $\max_\pi V^\pi(s)$ subject to $V_c^\pi(s) = 0$. This could be solved via a Lagrangian-based method:

$$\min_\pi \max_\lambda L(\pi, \lambda) = \min_\pi \max_\lambda \left( \mathbb{E}_{s \sim d_0} [-V^\pi(s) + \lambda V_c^\pi(s)] \right).$$

Now what remains is obtaining the reachability estimation function $\phi^*$. First, we address the problem of acquiring optimal likelihood of being feasible. It is nearly impossible to accurately know before training if a state is in a safest policy's infeasible set. We propose learning a function guaranteed to converge to this REF (with some discount factor for $\gamma$-contraction mapping) by using the recursive Bellman formulation proved in Theorem 1.

We learn a function $p(s)$ to capture the probability $\phi^*(s)$. It is trained like a reachability function:

$$p(s) = \max\{\mathbb{1}_{s \in S_v}, \gamma \cdot p(s')\},$$

where $S_v$ is the violation set, $s'$ is the next sampled state, and $\gamma$ is a discount parameter $0 \ll \gamma < 1$ to ensure convergence of $p(s)$. Furthermore, and crucially, we ensure the learning rate of this REF is on a slower time scale than the policy and its critics but faster than the lagrange multiplier.

Bringing the concepts covered above, we present our full optimization equation:

$$\min_\pi \max_\lambda L(\pi, \lambda) = \min_\pi \max_\lambda \left( \mathbb{E}_{s \sim d_0} \left[ [-V^\pi(s) + \lambda \cdot V_c^\pi(s)] \cdot (1 - p(s)) + V_c^\pi(s) \cdot p(s) \right] \right). \quad (4)$$

We show the design of our algorithm **RESPO** in an actor-critic framework in Algorithm 1. Note that the $V$ and $V_c$ have corresponding $Q$ functions: $V^\pi(s) = \mathbb{E}_{a \sim \pi(\cdot|s)} Q(s, a)$ and $V_c^\pi(s) = \mathbb{E}_{a \sim \pi(\cdot|s)} Q_c(s, a)$. The gradients' definitions are found in the appendix. We use operator $\Gamma_\Theta$ to indicate the projection of vector $\theta \in \mathbb{R}^n$ to the closest point in compact and convex set $\Theta \subseteq \mathbb{R}^n$. Specifically, $\Gamma_\Theta = \arg\min_{\hat\theta \in \Theta} ||\hat\theta - \theta||^2$. $\Gamma_\Omega$ is similarly defined.

---
**Algorithm 1** RESPO Actor Critic
---
**Require:** Randomly initialized policy $\pi_\theta$'s parameters $\theta_0$, reward critic $Q$'s parameters $\eta_0$, cost critic $Q_c$'s parameters $\kappa_0$, REF $p$'s parameters $\xi_0$, Lagrange multiplier $\lambda$'s parameters $\omega_0$, horizon $T$

**Require:** Convex projection operators $\Gamma_\Theta$ and $\Gamma_\Omega$, and reward and cost critic learning rate $\zeta_1(k)$, policy learning rate $\zeta_2(k)$, REF learning rate $\zeta_3(k)$, lagrange multiplier learning rate $\zeta_4(k)$

1: **for** $k = 0, 1, 2, ...$ **do**
2:   **for** $i = 0, 1, 2, ...$ **do**
3:     Sample trajectories $\tau_i : \{(s_j, a_j, s'_j, r_j, h_j)\} \sim \pi_\theta$
4:     **Rew. Update** $\eta_{k+1} = \eta_k - \zeta_1(k)\nabla_\eta Q(s_t, a_t) \cdot [Q(s_t, a_t) - (r(s_t, a_t) + \gamma Q(s_{t+1}, a_{t+1}))]$
5:     **Cost Update** $\kappa_{k+1} = \kappa_k - \zeta_1(k)\nabla_\kappa Q_c(s_t, a_t) \cdot [Q_c(s_t, a_t) - (h(s_t) + \gamma Q_c(s_{t+1}, a_{t+1}))]$
6:     **Policy Update** $\theta_{k+1} =$
7:     $\Gamma_\Theta\left(\theta_k - \zeta_2(k)\gamma^t\left[-Q(s_t, a_t)[1 - p(s_t)] + Q_c(s_t, a_t)[\lambda(1 - p(s_t)) + p(s_t)]\right]\nabla_\theta \log \pi_\theta(a_t|s_t)\right)$
8:     **REF Update** $\xi_{k+1} = \xi_k - \zeta_3(k)\nabla_\xi p(s_t) \cdot [p(s_t) - \max\{\mathbb{1}_{h(s_t)>0}, \gamma p(s_{t+1})\}]$
9:     **Lagrange multiplier Update** $\omega_{k+1} = \Gamma_\Omega\left(\omega_k - \zeta_4(k)Q_c(s_t, a_t)(1 - p(s_t))\nabla_\omega \lambda\right)$
10:   **end for**
11: **end for**
---

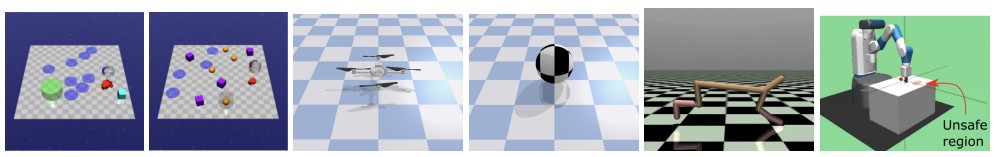

Figure 1: We compare the performance of our algorithm with other SOTA baselines in Safety Gym (left two figures), Safety PyBullet (middle two figures), and Safety MuJoCo (right two figures).

## 5.4 Convergence Analysis

We provide convergence analysis of our algorithm for Finite MDPs (finite bounded state and action space sizes, maximum horizon $T$, reward bounded by $R_{\max}$, and cost bounded by $H_{\max}$) under reasonable assumptions. We demonstrate our algorithm almost surely finds a locally optimal policy for our RESPO formulation, based on the following assumptions:

- **A1** *(Step size):* Step sizes follow schedules $\{\zeta_1(k)\}, \{\zeta_2(k)\}, \{\zeta_3(k)\}, \{\zeta_4(k)\}$ where:

$$\sum_k \zeta_i(k) = \infty \text{ and } \sum_k \zeta_i(k)^2 < \infty, \forall i \in \{1, 2, 3, 4\}, \quad \text{and} \quad \zeta_j(k) = o(\zeta_{j-1}(k)), \forall j \in \{2, 3, 4\}.$$

The reward returns and cost returns critic value functions must follow the fastest schedule $\zeta_1(k)$, the policy must follow the second fastest schedule $\zeta_2(k)$, the REF must follow the second slowest schedule $\zeta_3(k)$, and finally, the lagrange multiplier should follow the slowest schedule $\zeta_4(k)$.

- **A2** *(Strict Feasibility):* $\exists \pi(\cdot|\cdot; \theta)$ such that $\forall s \in \mathcal{S}_I$ where $\phi^*(s) = 0$, $V_c^{\pi_\theta}(s) \leq 0$.
- **A3** *(Differentiability and Lipschitz Continuity):* For all state-action pairs $(s, a)$, we assume value and cost Q functions $Q(s, a; \eta), Q_c(s, a; \kappa)$, policy $\pi(a|s; \theta)$, and REF $p(s, a; \xi)$ are continuously differentiable in $\eta, \kappa, \theta, \xi$ respectively. Furthermore, $\nabla_\omega \lambda_\omega$ and, for all state-action pairs $(s, a)$, $\nabla_\theta \pi(a|s; \theta)$ are Lipschitz continuous functions in $\omega$ and $\theta$ respectively.

The detailed proof of the following result is provided in the appendix.

**Theorem 2.** Given Assumptions **A1**-**A3**, the policy updates in Algorithm 1 will almost surely converge to a locally optimal policy for our proposed optimization in Equation RESPO.

## 6 Experiments

**Baselines.** The baselines we compare are CMDP-based or solve for hard constraints. The CMDP baselines are Lagrangian-based Proximal Policy Optimization (**PPOLag**) based on [7], Constraint-Rectified Policy Optimization (**CRPO**) [35], Penalized Proximal Policy Optimization (**P3O**) [32], and Projection-Based Constrained Policy Optimization (**PCPO**) [3]. The hard constraints baselines are **RCRL** [27], **CBF** with constraint $\dot{h}(s) + \nu \cdot h(s) \leq 0$, and Feasibile Actor-Critic (**FAC**) [9]. We classify **FAC** among the hard constraint approaches because we make its cost threshold $\chi = 0$ in order to better compare using NN lagrange multiplier with our REF approach in **RESPO**. We include the unconstrained Vanilla **PPO** [33] baseline for reference.

**Benchmarks.** We compare **RESPO** with the baselines in a diverse suite of safety environments. We consider high-dimensional environments in Safety Gym [30] (namely PointButton and CarGoal), Safety PyBullet [50] (namely DroneCircle and BallRun), and Safety MuJoCo [51], (namely Safety HalfCheetah and Reacher). We also show our algorithm in a multi-drone environment with *multiple hard and soft constraints*. More detailed experiment explanations and evaluations are in the appendix.

## 6.1 Main Experiments in Safety Gym, Safety PyBullet, and MuJoCo

We compare our algorithm with SOTA benchmarks on various high-dimensional (up to 76D observation space), complex environments in the stochastic setting, i.e., where the environment and/or policy are stochastic. Particularly, we examine environments in Safety Gym, Safety PyBullet, and Safety MuJoCo. The environments provide reward for achieving a goal behavior or location, while the cost is based on tangible (e.g., avoiding quickly moving objects) and non-tangible (e.g., satisfying speed limit) constraints. Environments like PointButton require intricate behavior where specific buttons must be reached while avoiding multiple moving obstacles, stationary hazards, and wrong buttons.

Overall, **RESPO** achieves the best balance between optimizing reward and minimizing cost violations across all the environments. Specifically, our approach generally has the highest reward performance (see the red lines from the top row of Figure 2) among the safety-constrained algorithms while maintaining reasonably low to 0 cost violations (like in HalfCheetah). When **RESPO** performs the second highest, the highest-performing safety algorithm always incurs several times more violations than **RESPO** – for instance, **RCRL** in PointButton or **PPOLag** in Drone Circle. Non-primal-dual CMDP approaches, namely **CRPO**, **P3O**, and **PCPO** generally satisfy their cost threshold constraints, but their reward performances rarely exceed that of **PPOLag**. **RCRL** generally has extremes of high reward and high cost, like in BallRun, or low reward and low cost, like in CarGoal. **FAC** and **CBF** generally have conservative behavior that sacrifices reward performance to minimize cost.

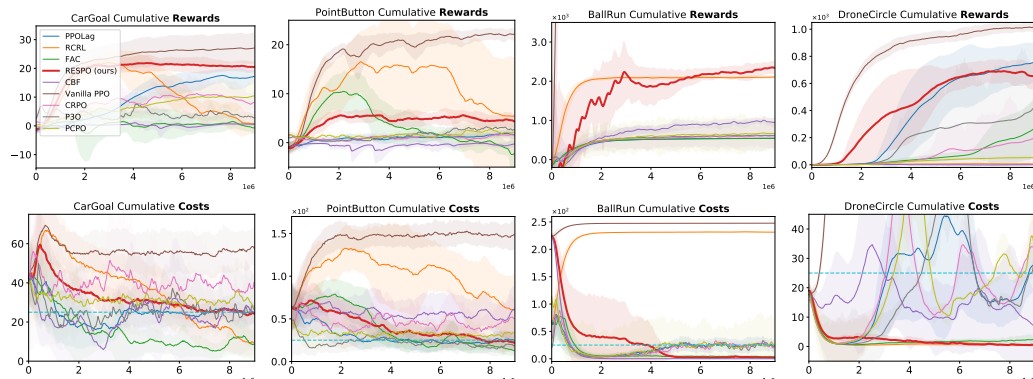

Figure 2: Comparison of RESPO with baselines in Safety Gym and PyBullet environments. The plots in the first row show performance measured in rewards (higher is better); those in second row show cost (lower is better). RESPO (red curves) achieves the best balance of maximizing reward and minimizing cost. When other methods achieve higher rewards than RESPO, they achieve much higher costs as well. E.g., in PointButton, RCRL has slightly higher rewards, but accumulates over $3\times$ violations than RESPO. Note, Vanilla PPO is unconstrained.

## 6.2 Hard and Soft Constraints

We also demonstrate **RESPO**'s performance in an environment with multiple hard and soft constraints. The environment requires controlling two drones to pass through a tunnel one at a time while respecting certain distance requirements. The reward is given for quickly reaching the goal positions. The two hard constraints involve **(H1)** ensuring neither drone collides into the wall and **(H2)** the distance between the two drones is more than 0.5 to ensure they do not collide. The soft constraint is that the two drones are within 0.8 of each other to ensure real-world communication. It is preferable to prioritize hard constraint **H1** over hard constraint **H2**, since colliding with the wall may have more serious consequences to the drones rather than violations of an overly precautious distance constraint.

Our approach, in the leftmost of Figure 3, successfully reaches the goal while avoiding the wall obstacles in all time steps. We are able to prioritize this wall avoidance constraint over the second hard constraint. This can be seen particularly in between the blue to cyan time period where the higher

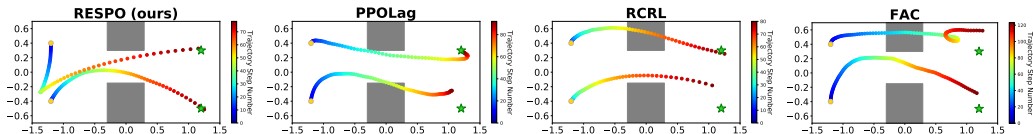

Figure 3: Comparison of RESPO with baselines trajectories in Hard & Soft Constraints multi-Drone control. Starting at gold circles, drones must enter the tunnel one at a time and reach green stars. Hard constraints are wall avoidance and ensuring drones are farther than $0.5$ meters from each other. Soft constraint is drones are within $0.8$ meters of each other. Trajectory colors correspond to time. RESPO (on left) successfully controls drones to reach goals while always avoiding walls and usually respecting distance constraints. Other baselines cannot manage multiple constraints: they collide with the wall and have many distance constraint violations.

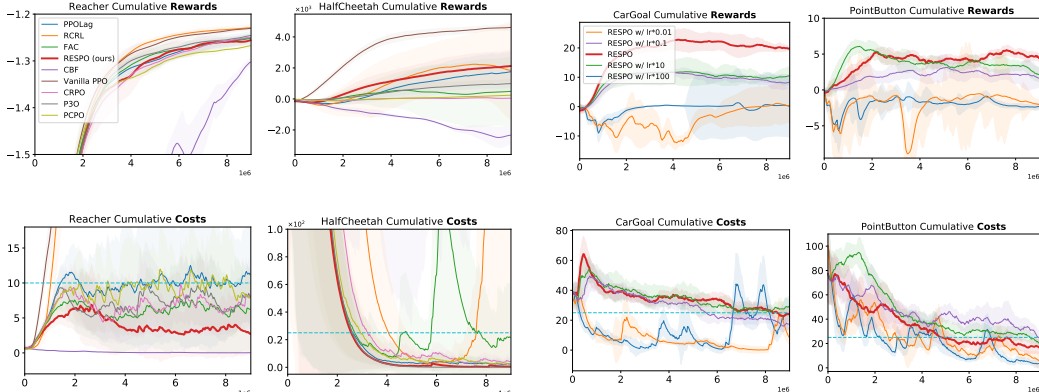

Figure 4: Comparison of RESPO with baselines in Mu-JoCo. Higher rewards (first row plots) and lower costs (second row plots) are better. In HalfCheetah, RESPO has highest reward among safety baselines, with $0$ violations. In Reacher, RESPO has good rewards, low costs.

Figure 5: Ablation study on the learning rate of REF. Higher rewards (first row plots) are better; lower costs (second row plots) are better. When changing REF's learning rate to violate timescale assumptions, REF produces suboptimal feasible sets.

Drone makes way for the lower Drone to pass through but needs to make a drop to make a concave parabolic trajectory to the goal. Nonetheless, the hard constraints are almost always satisfied, thereby producing the behavior of allowing one drone through the tunnel at a time. The soft constraints are satisfied at the beginning and end but are violated, reasonably, in the middle of the episode since only one drone can pass through the tunnel at a time, thereby forcing the other drone into a standby mode.

### 6.3 Ablation Studies

We also perform ablation studies to experimentally confirm the design choices we made based on the theoretically established convergence and optimization framework. We particularly investigate the effects of changing the learning rate of our reachability function as well as changing the optimization framework. We present the results of changing the learning rate for REF in Figure 5 while our results for the ablation studies on our optimization framework can be seen in Figure 6.

In Figure 5, we show the effects of making the learning rate of REF slower and faster than the one we use in accordance with Assumption 1. From these experiments, changing the learning rate in either direction produces poor reward performance. A fast learning rate makes the REF converge to the likelihood of infeasibility for the current policy, which can be suboptimal. But a very slow learning rate means the function takes too long to converge – the lagrange multiplier may meanwhile become very large, thus making it too difficult to optimize for reward returns. In both scenarios, the algorithm with modified learning rates produces conservative behavior that sacrifices reward performance.

In Figure 6, we compare **RESPO** with RCRL implemented with our REF and **PPOLag** in the CMDP framework with cost threshold $\chi = 0$ to ensure hard constraint satisfaction. The difference between **RESPO** and the RCRL-based ablation approach is that the ablation still uses $V_h^\pi$ instead of $V_c^\pi$. The ablation aproach's high cumulative cost can be attributed to the limitations of using $V_h^\pi$ – particularly, the lower sensitivity of $V_h^\pi$ to safety improvement and its lack of guarantees on feasible set (re)entrance. **PPOLag** with $\chi = 0$ produces low violations but also very low reward performance that's close to zero. Naively using $V_c^\pi$ in a hard constraints framework leads to very conservative

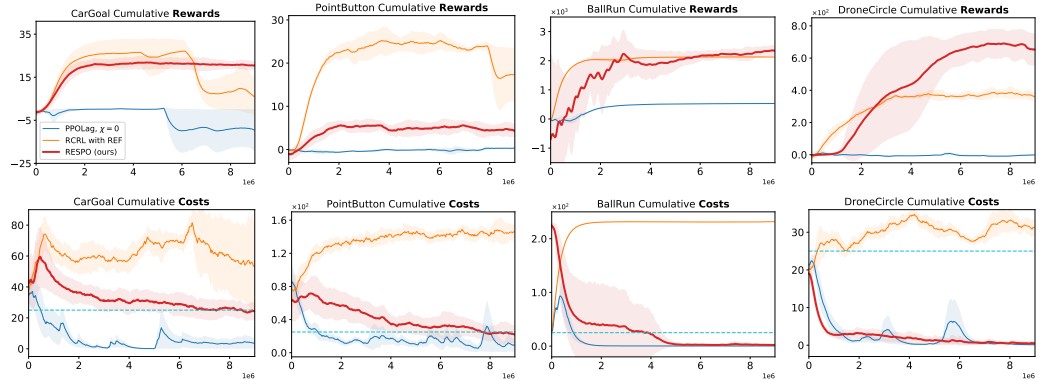

Figure 6: Ablation study on optimization framework. Top row plots show performance measured in reward (higher is better). Bottom row plots show cost (lower is better). RESPO (red curve) achieves best balance of maximizing reward and minimizing cost. RCRL framework implemented with our REF without $V_c^\pi$ incurs very high costs. PPOLag in CMDP framework with $\chi = 0$ has very low reward performance. So, learning REF and learning $V_c^\pi$ are both crucial components in our design and work in tandem to contribute to RESPO's efficacy.

behavior that sacrifices reward performance. Ultimately, this ablation study experimentally highlights the importance of learning our REF *and* using value function $V_c^\pi$ in our algorithm's design.

# 7 Discussion and Conclusion

In summary, we proposed a new optimization formulation and a class of algorithms for safety-constrained reinforcement learning. Our framework optimizes reward performance for states in least-violation policy's feasible state space while maintaining persistent safety as well as providing the safest behavior in other states by ensuring entrance into the feasible set with minimal cumulative discounted costs. Using our proposed reachability estimation function, we prove our algorithm's class of actor-critic methods converge a locally optimal policy for our proposed optimization. We provide extensive experimental results on a diverse suite of environments in Safety Gym, PyBullet, and MuJoCo, and an environment with multiple hard and soft constraints, to demonstrate the effectiveness of our algorithm when compared with several SOTA baselines. We leave open various extensions to our work to enable real-world deployment of our algorithm. These include constraining violations during training, guaranteeing safety in single-lifetime reinforcement learning, and ensuring policies don't forget feasible sets as environment tasks change. Our approach of learning the optimal REF to reduce the state space into a low-dimensional likelihood representation to guide training for high-dimensional policies can have applications in other learning problems in answering binary classification or likelihood-based questions about dynamics in high-dimension feature spaces.

# 8 Acknowledgements

This material is based on work supported by NSF Career CCF 2047034, NSF CCF DASS 2217723, ONR YIP N00014-22-1-2292, and Amazon Research Award.

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

# A  Notation

| | |
|---|---|
| $r$ | reward function |
| $h$ | safety loss function |
| $V^\pi$ | cumulative discounted reward value function |
| $V_c^\pi$ | cumulative discounted cost value function |
| $V_h^\pi$ | reachability value function |
| $\mathbb{1}_{s \in \mathcal{S}_v}$ | instantaneous violation indicator function |
| $\phi^\pi$ | reachability estimation function (REF) of a given policy |
| $\phi^*$ | reachability estimation function (REF) of safest policy |
| $p$ | predicted reachability estimation function (REF) of safest policy |
| $H_{\max}$ | upper bound of function $h$ |
| $H_{\min}$ | minimum non-zero value of of function $h$ |
| $\lambda_{\max}$ | maximum value to clip lagrange multiplier |
| $R_{\max}$ | upper bound of function $r$ |
| $\mathcal{S}_s$ | safe set |
| $\mathcal{S}_v$ | unsafe set |
| $\mathcal{S}_f$ | feasible set (persistent safe set) |
| $\mathbb{E}_{s' \sim \pi, P}$ | expectation taken over possible next states |
| $\mathbb{E}_{\tau \sim \pi, P}$ | expectation taken over possible trajectories |
| $\mathbb{E}_{s \sim d_0}$ | expectation taken over initial distribution |

Table 1: Notation used in the paper.

# B  Gradient estimates

The Q value losses based on the MSE between the Q networks and the respective sampled returns result in the gradients:

$$\hat{\nabla}_\eta J_Q(\eta) = \nabla_\eta Q(s_t, a_t; \eta) \cdot [Q_\eta(s_t, a_t) - (r(s_t, a_t) + \gamma Q(s_{t+1}, a_{t+1}; \eta))]$$

$$\hat{\nabla}_\kappa J_{Q_c}(\kappa) = \nabla_\kappa Q_c(s_t, a_t; \kappa) \cdot [Q_\kappa(s_t, a_t) - (h(s_t) + \gamma Q_c(s_{t+1}, a_{t+1}; \kappa))]$$

Similarly the REF gradient update is:

$$\hat{\nabla}_\eta J_p(\xi) = \nabla_\xi p(s_t; \xi) \cdot [p(s_t; \xi) - \max\{\mathbb{1}_{s_t \in S_v}, \gamma p(s_{t+1}; \xi)\}]$$

From the policy gradient theorem in [52], we get the policy gradient loss as:

$$\hat{\nabla}_\theta J_\pi(\theta) = \gamma^t \Bigg[ - Q_\eta(s_t, a_t)[1 - p_\xi(s_t)]$$
$$+ Q_c(s_t, a_t)[\lambda_\omega(1 - p_\xi(s_t)) + p_\xi(s_t)] \Bigg] \nabla_\theta \log \pi_\theta(a_t | s_t)$$

and the stochastic gradient of the multiplier is

$$\hat{\nabla}_\omega J_\lambda(\omega) = Q_c(s_t, a_t; \kappa)(1 - p_\xi(s_t))\nabla_\omega \lambda_\omega$$

and $\lambda_\omega$ is clipped to be in range $[0, \lambda_{\max}]$ (in particular, projection operator $\Gamma_\Omega(\lambda_\omega) = \arg\min_{\hat{\lambda}_\omega \in [0, \lambda_{\max}]} ||\lambda_\omega - \hat{\lambda}_\omega||^2$).

# C  Proofs

### C.1  Theorem 1 with Proof

**Theorem 3.** The REF can be reduced to the following recursive Bellman formulation:

$$\phi^\pi(s) = \max\{\mathbb{1}_{s \in \mathcal{S}_v}, \mathop{\mathbb{E}}_{s' \sim \pi, P(s)} \phi^\pi(s')\},$$

where $s' \sim \pi, P(s)$ is a sample of the immediate successive state (i.e., $s' \sim P(\cdot|s, a \sim \pi(\cdot|s))$) and the expectation is taken over all possible successive states.

*Proof.*

$$
\begin{aligned}
\phi^\pi(s) &:= \mathop{\mathbb{E}}_{\tau \sim \pi, P(s)} \max_{s_t \in \tau} \mathbb{1}_{s_t^\pi \in S_v} \\
&= \mathop{\mathbb{E}}_{\tau \sim \pi, P(s)} \max\{\mathbb{1}_{s \in S_v}, \max_{s_t \in \tau \setminus \{s\}} \mathbb{1}_{s_t^\pi \in S_v}\} \\
&= \max\{\mathbb{1}_{s \in S_v}, \mathop{\mathbb{E}}_{\tau \sim \pi, P(s)} \max_{s_t \in \tau \setminus \{s\}} \mathbb{1}_{s_t^\pi \in S_v}\} \\
&= \max\{\mathbb{1}_{s \in S_v}, \mathop{\mathbb{E}}_{s' \sim \pi, P(s)} \mathop{\mathbb{E}}_{\tau' \sim \pi, P(s')} \max_{s_t \in \tau'} \mathbb{1}_{s_t^\pi \in S_v}\} \\
&= \max\{\mathbb{1}_{s \in S_v}, \mathop{\mathbb{E}}_{s' \sim \pi, P(s)} \phi^\pi(s')\}
\end{aligned}
$$

Note that we use the notation $\tau \sim \pi, P(s)$ to indicate a trajectory sampled from the MDP with transition probability $P$ under policy $\pi$ starting from state $s$, and use the notation $s' \sim \pi, P(s)$ to indicate the next immediate state from the MDP with transition probability $P$ under policy $\pi$ starting from state $s$. The third line holds because the indicator function is either 0 or 1, so if it's 1 then $\phi^\pi(s) = \mathbb{E}_{\tau \sim \pi, P(s)} 1 = 1$ else $\phi^\pi(s) = \mathbb{E}_{\tau \sim \pi, P(s)} \max_{s_t \in \tau \setminus \{s\}} \mathbb{1}_{s_t^\pi \in S_v}$. $\qquad\square$

## C.2   Proposition 1 with Proof

**Proposition 3.** The cost value function $V_c^\pi(s)$ is zero for state $s$ if and only if the persistent safety is guaranteed for that state under the policy $\pi$.

*Proof.* (IF) Assume for a given policy $\pi$, the persistent safety is guaranteed, i.e. $h(s_t|s_0 = 0, \pi) = 0$ holds for all $s_t \in \tau$ for all possible trajectories $\tau$ sampled from the environment with control policy $\pi$. We then have:

$$V_c^\pi(s) := \mathop{\mathbb{E}}_{\tau \sim \pi, P(s)} \left[\sum_{s_t \in \tau} \gamma^t h(s_t)\right] = 0.$$

(ONLY IF) Assume for a given policy $\pi$, $V_c^\pi(s) = 0$. Since the image of the safety loss function $h(s)$ is non-negative real, and $V_c^\pi(s)$ is the expectation of the sum of non-negative real values, the only way $V_c^\pi(s) = 0$ is if $h(s_t|s_0 = 0, \pi) = 0, \forall s_t \in \tau$ for all possible trajectories $\tau$ sampled from the environment with control policy $\pi$. $\qquad\square$

## C.3   Proposition 2 with Proof

**Proposition 4.** If $\exists \pi$ that produces trajectory $\tau = \{(s_i), i \in \mathbb{N}, s_1 = s\}$ in deterministic MDP $\mathcal{M}$ starting from state $s$, and $\exists m \in \mathbb{N}, m < \infty$ such that $s_m \in S_f^\pi$, then $\exists \epsilon > 0$ where if discount factor $\gamma \in (1 - \epsilon, 1)$, then the optimal policy $\pi^*$ of Main paper Equation 3 will produce a trajectory $\tau' = \{(s'_j), j \in \mathbb{N}, s'_1 = s\}$, such that $\exists n \in \mathbb{N}, n < \infty, s'_n \in S_f^{\pi^*}$ and $V_c^{\pi^*}(s) = \min_{\pi'} V_c^{\pi'}(s)$.

In other words the proposition is stating for some state $s$, if there is a policy that enters its feasible set in a finite number $(m - 1)$ of steps, then by ensuring discount factor $\gamma$ is close to 1 we can guarantee that the optimal policy $\pi^*$ of Main paper Equation 3 will also enter the feasible set in a finite number of steps with the minimum cumulative discounted sum of the costs. Note that $\pi^*$ will always produce trajectories with the minimum discounted sum of costs whether the state is in the feasible or infeasible set of the policy by virtue of its optimization which constrains $V_c^\pi$.

*Proof.* We consider two cases: (Case 1) $m = 1$ and (Case 2) $m > 1$.

**Case 1** $m = 1$**:** In this case, there exists a policy $\pi$ in which the the current state $s$ is in the feasible set of that policy. By definition, that means that in a trajectory $\tau$ sampled in the MDP using that

policy, starting from state $s$, there are no future violations incurred in $\tau$. Thus $V_c^\pi(s) = 0$. Since $\pi^*$ incurs the minimum cumulative violation, $V_c^{\pi^*}(s) = 0$ trivially. Therefore, $s$, the first state of the trajectory, is in the feasible set of $\pi^*$.

**Case 2** $m > 1$**:** Since policy $\pi^*$ produces the minimum cumulative discounted cost for a given state $s$, the core of this proof will be demonstrating that the minimum cumulative discounted cost of *entering* the feasible set (call this value $H_E$) is less than the minimum cumulative discounted cost of *not entering* the feasible set (call this value $H_N$), and therefore $\pi^*$ will choose the route of entering the feasible set.

The proof will proceed by deriving a sufficient condition for $H_E < H_N$ by establishing bounds on them.

We place an upper bound on the minimum cumulative discounted cost of entering the feasible set $H_E$. Since $\exists \pi$ that enters the feasible set in $m-1$ steps, entering the feasible set can be at most the highest possible cost that $\pi$ incurs. Since the maximum cost at any state is $H_{\max}$, the upper bound is the discounted sum of $m-1$ steps of violations $H_{\max}$, or

$$H_E < \frac{H_{\max}(1 - \gamma^{m-1})}{(1 - \gamma)}$$

We place a lower bound on the minimum cumulative discounted cost of not entering the feasible set $H_N$. In this case, say in the sampled trajectory, the maximum gap between any two non-zero violations is $w$. By definition, the trajectory cannot have an infinite sequence of violation-free states since the trajectory never enters the feasible set. Therefore $w$ is finite. Now recall $H_{\min}$ is the lower bound on the non-zero values of $h$. So the minimum cumulative discounted cost of not entering the feasible set must be at least the cost of the trajectory with a violation of $H_{\min}$ at intervals of $w$ steps. That is:

$$\frac{H_{\min}(\gamma^w)}{(1 - \gamma^w)} < H_N$$

Now $H_E < H_N$ will be true if the upper bound of $H_E$ is less than the lower bound of $H_N$. In other words $H_E < H_N$ is true if:

$$\frac{H_{\max}(1 - \gamma^{m-1})}{(1 - \gamma)} < \frac{H_{\min}(\gamma^w)}{(1 - \gamma^w)} \tag{5}$$

Rearranging, we get:

$$\frac{H_{\max}}{H_{\min}} < \frac{(1 - \gamma) \cdot (\gamma^w)}{(1 - \gamma^{m-1}) \cdot (1 - \gamma^w)} \tag{6}$$

Let's define the RHS of the Inequality 6 as the function $\upsilon(\gamma)$. Consider $\gamma \in (0, 1)$. It is not difficult to demonstrate that $\upsilon(\gamma)$ in this domain range is a continuous function and that left directional limit $\lim_{\gamma \to 1^-} \upsilon(\gamma) = \infty$. This suggests that there is an open interval of values for $\gamma$ (whose supremum is 1) for which $H_{\max}/H_{\min} < \upsilon(\gamma)$ and so $H_E < H_N$. So we establish that $\exists \epsilon > 0$ such that for $\gamma \in (1 - \epsilon, 1)$, we satisfy the sufficient condition $H_E < H_N$ so that the optimal policy will enter its feasible set.

Thus, we prove that if there is a policy entering its feasible set from state $s$, then there is a range of values for $\gamma$ that are close enough to 1 ensuring that the optimal policy of Main paper Equation 3 will enter its feasible set in a finite number of steps with minimum discounted sum of costs.

$\square$

### C.4  Theorem 2 with Proof

**Theorem 4.** Given Assumptions **A1**-**A3** in Main paper, the policy updates in Algorithm 1 will almost surely converge to a locally optimal policy for our proposed optimization in Equation RESPO.

We first provide an intuitive explanation behind why our REF learns to converge to the safest policy's REF, then a proof overview, and then the full proof.

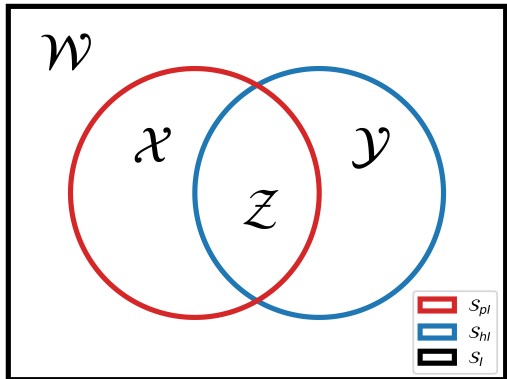

Figure 7: The predicted feasible set converges to a safest policy's feasible set since the misclassified regions $\mathcal{X}$ and $\mathcal{Y}$ are corrected over time.

### C.4.1 Intuition behind REF convergence

The approach can be explained by considering what happens in the individual regions of space. Consider a deterministic environment for simplicity. As seen in Figure 7, there are two subsets of the initial state space: a safest policy's "true" feasible set $\mathcal{S}_{hI}$ and REF predicted feasible set $\mathcal{S}_{pI}$, and they create 4 regions in the initial state space $\mathcal{S}_I$: $\mathcal{W} = \overline{\mathcal{S}_{hI}} \cap \overline{\mathcal{S}_{pI}}$, $\mathcal{X} = \overline{\mathcal{S}_{hI}} \cap \mathcal{S}_{pI}$, $\mathcal{Y} = \mathcal{S}_{hI} \cap \overline{\mathcal{S}_{pI}}$, $\mathcal{Z} = \mathcal{S}_{hI} \cap \mathcal{S}_{pI}$. Consider a point during training when the lagrange multiplier $\lambda$ is sufficiently large. For states in $\mathcal{W}$, the set of correctly classified infeasible states, the algorithm will simply minimize cumulative violations $V_c^{\pi_\theta}(s)$, and thereby remain as safe as possible since the policy and critics learning rates are faster than that of REF. $\mathcal{X}$, which is the set of infeasible states that are misclassified, is very small if we ensure the policy and REF are trained at much faster time scales than the multiplier and so when the agent starts in true infeasible states, it will by definition reach violations and therefore be labeled as infeasible. In $\mathcal{Y}$, the set of truly feasible states that are misclassified, the algorithm also minimizes cumulative violations, which by the definition of feasibility should be 0. It will then have no violations and enter the correctly predicted feasible set $\mathcal{Z}$. And when starting in states in $\mathcal{Z}$, the algorithm will optimize the lagrangian, and since the multiplier $\lambda$ is sufficiently large, it will converge to a policy that optimizes for reward while ensuring safety, i.e. no future violations, and therefore the state will stay predictably feasible in $\mathcal{Z}$. In this manner, REF's predicted feasible set will converge to the optimal feasible set, and the agent will be safe and have optimal performance in the feasible set and be the safest behavior outside the feasible set. Thereby, the algorithm finds a locally optimal solution to the proposed optimization formulation.

### C.4.2 Proof Overview

We show our algorithm convergence to the optimal policy by utilizing the proof framework of multi-time scale presented in [8, 10, 53, 27]. Specifically, we have 4 time scales for (1) the critics, (2) policy, (3) REF function, and (4) lagrange multiplier, listed in order from fastest to slowest. The overview of each timescale proof step is as follows:

1 We demonstrate the almost sure convergence of the critics to the corresponding fixed point optimal critic functions of the policy.

2 Using multi-timescale theory, we demonstrate the policy almost surely converges to a stationary point of a continuous time system, which we show has a Lyapunov function certifying its locally asymptotic stability at the stationary point.

3 We demonstrate the almost sure convergence of the REF function to the REF of the policy that is safe insofar as the lagrange multiplier is sufficiently large.

4 We demonstrate the almost sure convergence of the lagrange multiplier to a stationary point similar to the proof in the policy timecale.

Finally, we demonstrate that the stationary points for the policy and lagrange multiplier form a saddle point, and so by local saddle point theorem we almost surely achieve the locally optimal policy of our proposed optimization.

### C.4.3 Proof Details

*Proof.* **Step 1** (convergence of the critics $V_\eta$ and $V_\kappa$ updates): From the multi-time scale assumption, we know that $\eta$ and $\kappa$ will convergence on a faster time scale than the other parameters $\theta$, $\xi$, and $\omega$. Therefore, we can leverage Lemma 1 of Chapter 6 of [10] to analyze the convergence properties while updating $\eta_k$ and $\kappa_k$ by treating $\theta$, $\xi$, and $\omega$ as fixed parameters $\theta_k$, $\xi_k$, and $\omega_k$. In other words, the policy, REF, and lagrange multiplier are fixed while computing $Q^{\pi_{\theta_k}}(s, a)$ and $Q_c^{\pi_{\theta_k}}(s, a)$. With the Finite MDP assumption and policy evaluation convergence results of [52], and assuming sufficiently expressive function approximator (i.e. wide enough neural networks) to ensure convergence to global mininum, we can use the fact that the bellman operators $\mathcal{B}$ and $\mathcal{B}_c$ which are defined as

$$\mathcal{B}[Q](s, a) = r(s, a) + \gamma \underset{s', a' \sim \pi, P(s)}{\mathbb{E}}[Q(s', a')]$$

$$\mathcal{B}[Q_c](s, a) = h(s) + \gamma \underset{s', a' \sim \pi, P(s)}{\mathbb{E}}[Q_c(s', a')]$$

are $\gamma$-contraction mappings, and therefore as $k$ approaches $\infty$, we can be sure that $Q(s, a; \eta_k) \to Q(s, a; \eta^*) = Q^{\pi_{\theta_k}}(s, a)$ and $Q_c(s, a; \kappa_k) \to Q_c(s, a; \kappa^*) = Q_c^{\pi_{\theta_k}}(s, a)$. So since $\eta_k$ and $\kappa_k$ converge to $\eta^*$ and $\kappa^*$, we prove convergence of the critics in Time scale 1.

**Step 2** (convergence of the policy $\pi_\theta$ update): Because $\xi$ and $\omega$ updated on slower time scales than $\theta$, we can again use Lemma 1 of Chapter 6 of [10] and treat these parameters are fixed at $\xi_k$ and $\omega_k$ respectively when updating $\theta_k$. Additionally in Time scale 2, we have $||Q(s, a; \eta_k) - Q(s, a; \eta^*)|| \to 0$ and $||Q_c(s, a; \kappa_k) - Q_c(s, a; \kappa^*)|| \to 0$ almost surely. Now the update of the policy $\theta$ using the gradient from Equation 4 is:

$$\begin{aligned}
\theta_{k+1} &= \Gamma_\Theta[\theta_k - \zeta_2(k)(\nabla_\theta L(\theta, \xi_k, \omega_k)|_{\theta=\theta_k})] \\
&= \Gamma_\Theta[\theta_k - \zeta_2(k)[\gamma^t[-Q_\eta(s_t, a_t)[1 - p_{\xi_k}(s_t)] \\
&\quad + Q_c(s_t, a_t)[\lambda_\omega(1 - p_{\xi_k}(s_t)) + p_{\xi_k}(s_t)]]\nabla_\theta \log \pi(a_t|s_t; \theta)|_{\theta=\theta_k}]] \\
&= \Gamma_\Theta[\theta_k - \zeta_2(k)(\nabla_\theta L(\theta, \xi_k, \omega_k)|_{\theta=\theta_k, \eta=\eta^*, \kappa=\kappa^*} + \delta\theta_{k+1} + \delta\theta_\epsilon)]
\end{aligned}$$

where

$$\begin{aligned}
\delta\theta_{k+1} = \sum_{s_i, a_i} &\left[ d_0(s_0)P^{\pi_{\theta_k}}(s_i, a_i|s_0)\gamma^i[-Q_\eta(s_i, a_i)[1 - p_{\xi_k}(s_i)] \right. \\
&\left. + Q_c(s_i, a_i)[\lambda_\omega(1 - p_{\xi_k}(s_i)) + p_{\xi_k}(s_i)]]\nabla_\theta \log \pi(a_i|s_i; \theta)|_{\theta=\theta_k} \right] \\
&- \gamma^t[-Q_\eta(s_t, a_t)[1 - p_{\xi_k}(s_t)] + Q_c(s_t, a_t)[\lambda_\omega(1 - p_{\xi_k}(s_t)) + p_{\xi_k}(s_t)]] \\
&\qquad\qquad\qquad\qquad\qquad\qquad\qquad \cdot \nabla_\theta \log \pi(a_t|s_t; \theta)|_{\theta=\theta_k}
\end{aligned}$$

and

$$\begin{aligned}
\delta\theta_\epsilon = \sum_{s_i, a_i} &d_0(s_0)P^{\pi_{\theta_k}}(s_i, a_i|s_0)\left[\vphantom{\sum}\right. \\
-\gamma^i[-Q(s_i, a_i; \eta_k)[1 - p_{\xi_k}(s_i)] &+ Q_c(s_i, a_i; \kappa_k)[\lambda_\omega(1 - p_{\xi_k}(s_i)) + p_{\xi_k}(s_i)]] \\
&\qquad\qquad\qquad\qquad\qquad \cdot \nabla_\theta \log \pi(a_i|s_i; \theta)|_{\theta=\theta_k} \\
+\gamma^i[-Q^{\pi_{\theta_k}}(s_i, a_i)[1 - p_{\xi_k}(s_i)] &+ Q_c^{\pi_{\theta_k}}(s_i, a_i)[\lambda_\omega(1 - p_{\xi_k}(s_i)) + p_{\xi_k}(s_i)]] \\
&\qquad\qquad\qquad\qquad\left. \cdot \nabla_\theta \log \pi(a_i|s_i; \theta)|_{\theta=\theta_k}\right]
\end{aligned}$$

*Lemma 1*: We can first demonstrate that $\delta\theta_{k+1}$ is square integrable. In particular,

$$\mathbb{E}[||\delta\theta_{k+1}||^2|\mathcal{F}_{\theta,k}]$$

$$\leq 2||\nabla_\theta \log \pi(a|s;\theta)|_{\theta=\theta_k} \mathbb{1}_{\pi(a|s;\theta_k)>0}||^2_\infty \cdot \Big( ||Q(s,a;\eta_k)||^2_\infty \cdot ||1-p_{\xi_k}(s)||^2_\infty$$

$$+ ||Q_c(s,a;\kappa_k)||^2_\infty \cdot \Big[ ||\lambda_\omega||^2_\infty \cdot ||1-p_{\xi_k}(s)||^2_\infty + ||p_{\xi_k}(s)||^2_\infty \Big] \Big)$$

$$\leq 2\frac{||\nabla_\theta \pi(a|s;\theta)|_{\theta=\theta_k}||^2_\infty}{\min\{\pi(a|s;\theta_k)|\pi(a|s;\theta_k)>0\}} \cdot \Big( ||Q(s,a;\eta_k)||^2_\infty \cdot ||1-p_{\xi_k}(s)||^2_\infty$$

$$+ ||Q_c(s,a;\kappa_k)||^2_\infty \cdot \Big[ ||\lambda_\omega||^2_\infty \cdot ||1-p_{\xi_k}(s)||^2_\infty + ||p_{\xi_k}(s)||^2_\infty \Big] \Big)$$

Note that $\mathcal{F}_{\theta,k} = \sigma(\theta_m, \delta\theta_m, m \leq k)$ is the filtration for $\theta_k$ generated by different independent trajectories [8]. Also note that the indicator function is used because the expectation of $||\delta\theta_{k+1}||^2$ is taken with respect to $P^{\pi_{\theta_k}}$ and $P^{\pi_{\theta_k}}(s,a|s_0) = 0$ if $\pi(a|s;\theta_k) = 0$. From the Assumptions on Lipschitz continuity and Finite MDPs reward and costs, we can bound the values of the functions and the gradients of functions. Specifically

$$||\nabla_\theta \pi(a|s;\theta)|_{\theta=\theta_k}||^2_\infty \leq K_1(1+||\theta_k||^2_\infty),$$

$$||Q(s,a;\eta_k)||^2_\infty \leq \frac{R_{\max}}{1-\gamma},$$

$$||Q_h(s,a;\kappa_k)||^2_\infty \leq \frac{H_{\max}}{1-\gamma},$$

$$||\lambda_\omega||^2_\infty \leq \lambda_{\max},$$

$$||1-p_{\xi_k}(s)||^2_\infty \leq 1,$$

$$||p_{\xi_k}(s)||^2_\infty \leq 1$$

where $K_1$ is a Lipschitz constant. Furthermore, note that because we are sampling, $\pi(a|s;\theta_k)$ will take on only a finite number of values, so its nonzero values will be bounded away from zero. Thus we can say

$$\frac{1}{\min\{\pi(a|s;\theta_k)|\pi(a|s;\theta_k)>0\}} \leq K_2$$

for some large enough $K_2$. Thus using the bounds from these conditions, we can demonstrate

$$\mathbb{E}[||\delta\theta_{k+1}||^2|\mathcal{F}_{\theta,k}] \leq 2 \cdot K_1(1+||\theta_k||^2_\infty) \cdot K_2(\frac{R_{\max}}{1-\gamma} \cdot 1 + \frac{H_{\max}}{1-\gamma} \cdot (\lambda_{\max} \cdot 1 + 1)) < \infty$$

Therefore $\delta\theta_{k+1}$ is square integrable.

*Lemma 2*: Secondly, we can demonstrate $\delta\theta_\epsilon \to 0$.

$$\delta\theta_\epsilon = \sum_{s_i,a_i} d_0(s_0)P^{\pi_{\theta_k}}(s_i,a_i|s_0)\Big[ \gamma^i \big[(Q(s_i,a_i;\eta_k) - Q^{\pi_{\theta_k}}(s_i))[1-p_{\xi_k}(s_i)]$$

$$+ (-Q_c(s_i,a_i;\kappa_k) + Q_c^{\pi_{\theta_k}}(s_i))[\lambda_\omega(1-p_{\xi_k}(s_i)) + p_{\xi_k}(s_i)]]\nabla_\theta \log \pi(a_i|s_i;\theta)|_{\theta=\theta_k}\Big]$$

$$\leq \sum_{s_i,a_i} d_0(s_0)P^{\pi_{\theta_k}}(s_i,a_i|s_0)\Big[ \gamma^i \big[(Q(s_i,a_i;\eta_k) - Q(s_i,a_i;\eta^*))[1-p_{\xi_k}(s_i)]$$

$$+ (-Q_c(s_i,a_i;\kappa_k) + Q_c(s_i,a_i;\kappa^*))[\lambda_\omega(1-p_{\xi_k}(s_i)) + p_{\xi_k}(s_i)]]\nabla_\theta \log \pi(a_i|s_i;\theta)|_{\theta=\theta_k}\Big]$$

$$\leq \sum_{s_i,a_i} d_0(s_0)P^{\pi_{\theta_k}}(s_i,a_i|s_0)\Big[ \gamma^i \big[||Q(s_i,a_i;\eta_k) - Q(s_i,a_i;\eta^*)||[1-p_{\xi_k}(s_i)]$$

$$+ ||-Q_c(s_i,a_i;\kappa_k) + Q_c(s_i,a_i;\kappa^*)||[\lambda_\omega(1-p_{\xi_k}(s_i)) + p_{\xi_k}(s_i)]]\nabla_\theta \log \pi(a_i|s_i;\theta)|_{\theta=\theta_k}\Big]$$

And because we have $||Q(s,a;\eta_k) - Q(s,a;\eta^*)|| \to 0$ and $||Q_c(s,a;\kappa_k) - Q_c(s,a;\kappa^*)|| \to 0$ almost surely, we can therefore say $\delta\theta_\epsilon \to 0$.

*Lemma 3*: Finally, since $\hat{\nabla}_\theta J_\pi(\theta)|_{\theta=\theta_k}$ is a sample of $\nabla_\theta L(\theta,\xi_k,\omega_k)|_{\theta=\theta_k}$ based on the history of sampled trajectories, we conclude that $\mathbb{E}[\delta\theta_{k+1}|\mathcal{F}_{\theta,k}] = 0$.

From the 3 above lemmas, the policy $\theta$ update is a stochastic approximation of a continuous system $\theta(t)$ defined by [10]

$$\dot{\theta} = \Upsilon_\Theta[-\nabla_\theta L(\theta,\xi,\omega)] \tag{7}$$

in which

$$\Upsilon_\Theta[M(\theta)] \triangleq \lim_{0<\psi\to 0} \frac{\Gamma_\Theta(\theta + \psi M(\theta)) - \Gamma_\Theta(\theta)}{\psi}$$

or in other words the left directional derivative of $\Gamma_\Theta(\theta)$ in the direction of $M(\theta)$. Using the left directional derivative $\Upsilon_\Theta[-\nabla_\theta L(\theta,\xi,\omega)]$ in the gradient descent algorithm for learning the policy $\pi_\theta$ ensures the gradient will point in the descent direction along the boundary of $\Theta$ when the $\theta$ update hits its boundary. Using Step 2 in Appendix A.2 from [8], we have that $dL(\theta,\xi,\omega)/dt = -\nabla_\theta L(\theta,\xi,\omega)^T \cdot \Upsilon_\Theta[-\nabla_\theta L(\theta,\xi,\omega)] \leq 0$ and the value is non-zero if $||\Upsilon_\Theta[-\nabla_\theta L(\theta,\xi,\omega)]|| \neq 0$. Now consider the continuous system $\theta(t)$. For some fixed $\xi$ and $\omega$, define a Lyapunov function

$$\mathcal{L}_{\xi,\omega}(\theta) = L(\theta,\xi,\omega) - L(\theta^*,\xi,\omega)$$

where $\theta^*$ is a local minimum point. Then there exists a ball centered at $\theta^*$ with a radius $\rho$ such that $\forall\theta \in \mathfrak{B}_{\theta^*}(\rho) = \{\theta|||\theta-\theta^*|| \leq \rho\}$, $\mathcal{L}_{\xi,\omega}(\theta)$ is a locally positive definite function, that is $\mathcal{L}_{\xi,\omega}(\theta) \geq 0$. Using Proposition 1.1.1 from [54], we can show that $\Upsilon_\Theta[-\nabla_\theta L(\theta,\xi,\omega)]|_{\theta=\theta^*} = 0$ meaning $\theta^*$ is a stationary point. Since $dL(\theta,\xi,\omega)/dt \leq 0$, through Lyapunov theory for asymptotically stable systems presented in Chapter 4 of [55], we can use the above arguments to demonstrate that with any initial conditions of $\theta(0) \in \mathfrak{B}_{\theta^*}(\rho)$, the continuous state trajectory of $\theta(t)$ converges to $\theta^*$. Particularly, $L(\theta^*,\xi,\omega) \leq L(\theta(t),\xi,\omega) \leq L(\theta(0),\xi,\omega)$ for all $t > 0$.

Using these aforementioned properties, as well as the facts that 1) $\nabla_\theta L(\theta,\xi,\omega)$ is a Lipschitz function (using Proposition 17 from [8]), 2) the step-sizes of Assumption on steps sizes, 3) $\delta\theta_{k+1}$ is a square integrable Martingale difference sequence and $\delta\theta_\epsilon$ is a vanishing error almost surely, and 4) $\theta_k \in \Theta, \forall k$ implying that $\sup_k ||\theta_k|| < \infty$ almost surely, we can invoke Theorem 2 of chapter 6 in [10] to demonstrate the sequence $\{\theta_k\}, \theta_k \in \Theta$ converges almost surely to the solution of the ODE defined by Equation 7, which additionally converges almost surely to the local minimum $\theta^* \in \Theta$.

**Step 3** (convergence of REF $p_\xi$ updates): Since $\omega$ is updated on a slower time scale that $\xi$, we can again treat $\omega$ as a fixed parameter at $\omega_k$ when updating $\xi$. Furthermore, in Time scale 3, we know that the policy has converged to a local minimum, particularly $||\theta_k - \theta^*(\xi_k,\omega_k)|| = 0$. Now the bellman operator for REF is defined by

$$\mathcal{B}_p[p](s) = \max\{\mathbb{1}_{s\in S_v}, \gamma \mathbb{E}_{s'\sim\pi,P(s)}[p(s')]\}.$$

We demonstrate this is a $\gamma$ contraction mapping as follows:

$$|\mathcal{B}_p[p](s) - \mathcal{B}_p[\hat{p}](s)|$$
$$= |\max\{\mathbb{1}_{s\in S_v}, \gamma \mathbb{E}_{s'\sim\pi,P(s)}[p(s')]\} - \max\{\mathbb{1}_{s\in S_v}, \gamma \mathbb{E}_{s'\sim\pi,P(s)}[\hat{p}(s')]\}|$$
$$\leq |\gamma \mathbb{E}_{s'\sim\pi,P(s)}[p(s')] - \gamma \mathbb{E}_{s'\sim\pi,P(s)}[\hat{p}(s')]|$$
$$= \gamma|\mathbb{E}_{s'\sim\pi,P(s)}[p(s') - \hat{p}(s')]|$$
$$\leq \gamma \sup_s |p(s) - \hat{p}(s)| = \gamma||p - \hat{p}||_\infty$$

So we can say that $p(s;\xi_k)$ will converge to $p(s;\xi^*)$ as $k \to \infty$ under the same assumptions of the Finite MDP and function approximator expressiveness in Step 1. Therefore, $\pi_{\theta_k}$ will also converge to $\pi^\diamond = \pi_{\theta^*(\xi^*,\omega_k)}$ as $k \to \infty$. And because $\pi_\theta$ is the sampling policy used to compute $p$, $p(s;\xi^*) = p^{\pi_{\theta^*(\xi^*,\omega_k)}}(s;\xi^*) = p^\diamond(s)$.

Notice that $\pi^\diamond$ is a locally minimum optimal policy for the following optimization (recall $\lambda_\omega$ is treated as constant in this timescale):

$$\min_\pi \mathbb{E}_{s\sim d_0} \mathbb{E}_{a\sim\pi(\cdot|s)} \left[ -Q^\pi(s,a)\cdot[1-p^\diamond(s)] + Q_c^\pi(s,a)\cdot[(1-p^\diamond(s))\lambda_\omega + p^\diamond(s)] \right]$$

and therefore also locally minimum optimal policy for optimization:

$$\min_\pi \mathbb{E}_{s\sim d_0} \mathbb{E}_{a\sim\pi(\cdot|s)} \left[ -Q^\pi(s,a) + Q_c^\pi(s,a)\cdot[\lambda_\omega + \frac{p^\diamond(s)}{(1-p^\diamond(s))}] \right], \text{ if } p^\diamond(s) > 0$$

$$\mathbb{E}_{s\sim d_0} \min_\pi \mathbb{E}_{a\sim\pi(\cdot|s)} \left[ Q_c^\pi(s,a) \right], \text{ if } p^\diamond(s) = 0$$

Since $\frac{p^\diamond(s)}{(1-p^\diamond(s))} \geq 0$, and the $Q$ functions are always nonnegative, we can know that $\pi^\diamond$ is at least as safe as (i.e., its expected cumulative cost is at most that of) a locally optimal policy for the optimization:

$$\min_\pi \mathbb{E}_{s\sim d_0} \mathbb{E}_{a\sim\pi(\cdot|s)} \left[ -Q^\pi(s,a) + Q_c^\pi(s,a)\lambda_\omega \right] \tag{8}$$

As $\lambda_\omega$ approaches $\lambda_{\max}$, which in turn approaches $\infty$, the local minimum optimal policies of Equation 8 approach those of the optimization $\pi^\triangle = \arg\min_\pi \mathbb{E}_{s\sim d_0} \mathbb{E}_{a\sim\pi(\cdot|s)} Q_c^\pi(s,a)\lambda_\omega = \arg\min_\pi \mathbb{E}_{s\sim d_0} \mathbb{E}_{a\sim\pi(\cdot|s)} Q_c^\pi(s,a)$. Therefore, the feasible set of the REF $p^\diamond$ will approach that of the REF $p^{\pi^\triangle}$.

**Step 4** (convergence of lagrange multiplier $\lambda_\omega$ update): Since $\lambda_\omega$ is on the slowest time scale, we have that $||\theta_k - \theta^*(\omega)|| = 0$, $||\xi_k - \xi^*(\omega)|| = 0$, and $||Q_c(s,a;\kappa_k) - Q_c^{\pi_{\theta_k}}(s,a)|| = 0$ almost surely. Furthermore, due to the continuity of $\nabla_\omega L(\theta,\xi,\omega)$, we have that $||\nabla_\omega L(\theta,\xi,\omega)|_{\theta=\theta_k,\xi=\xi_k,\omega=\omega_k} - \nabla_\omega L(\theta,\xi,\omega)|_{\theta=\theta^*(\omega_k),\xi=\xi^*(\omega_k),\omega=\omega_k}|| = 0$ almost surely. The update of the multiplier using the gradient for Equation is:

$$\omega_{k+1} = \Gamma_\Omega[\omega_k + \zeta_4(k)(\nabla_\omega L(\theta,\xi,\omega)|_{\theta=\theta_k,\xi=\xi_k,\omega=\omega_k})]$$
$$= \Gamma_\Omega[\omega_k + \zeta_4(k)(Q_c(s_t,a_t;\kappa_k)[1-p(s_t;\xi_k)]\nabla_\omega\lambda_\omega|_{\omega=\omega_k})]$$
$$= \Gamma_\Omega[\omega_k + \zeta_4(k)(\nabla_\omega L(\theta,\xi,\omega)|_{\theta=\theta^*(\omega_k),\xi=\xi^*(\omega_k),\omega=\omega_k} + \delta\omega_{k+1})]$$

where

$$\delta\omega_{k+1} = -\nabla_\omega L(\theta,\xi,\omega)|_{\theta=\theta^*(\omega_k),\xi=\xi^*(\omega_k),\omega=\omega_k} + Q_c(s_t,a_t;\kappa_k)[1-p(s_t;\xi_k)]\nabla_\omega\lambda_\omega|_{\omega=\omega_k}$$

$$= -\sum_{s_i,a_i} d_0(s_0)P^{\pi_{\theta_k}}(s_i,a_i|s_0)[Q_c^{\pi_{\theta^*}}(s_i,a_i)[1-p_{\xi^*}(s_i)]\nabla_\omega\lambda_\omega|_{\omega=\omega_k}]$$
$$+ Q_c(s_t,a_t;\kappa_k)[1-p(s_t;\xi_k)]\nabla_\omega\lambda_\omega|_{\omega=\omega_k}$$

$$= -\sum_{s_i,a_i} d_0(s_0)P^{\pi_{\theta_k}}(s_i,a_i|s_0)[Q_c^{\pi_{\theta^*}}(s_i,a_i)[1-p_{\xi^*}(s_i)]\nabla_\omega\lambda_\omega|_{\omega=\omega_k}]$$
$$+ [Q_c(s_t,a_t;\kappa_k)[1-p(s_t;\xi_k)] - Q_c^{\pi_{\theta_k}}(s_t,a_t)[1-p(s_t;\xi_k)]+$$
$$Q_c^{\pi_{\theta_k}}(s_t,a_t)[1-p(s_t;\xi_k)] - Q_c^{\pi_{\theta_k}}(s_t,a_t)[1-p^\diamond(s_t)]+$$
$$Q_c^{\pi_{\theta_k}}(s_t,a_t)[1-p^\diamond(s_t)]]\nabla_\omega\lambda_\omega|_{\omega=\omega_k}$$

$$= -\sum_{s_i,a_i} d_0(s_0)P^{\pi_{\theta_k}}(s_i,a_i|s_0)[Q_c^{\pi_{\theta^*}}(s_i,a_i)[1-p_{\xi^*}(s_i)]\nabla_\omega\lambda_\omega|_{\omega=\omega_k}]$$
$$+ [(Q_c(s_t,a_t;\kappa_k) - Q_c^{\pi_{\theta_k}}(s_t,a_t))[1-p(s_t;\xi_k)]+$$
$$Q_c^{\pi_{\theta_k}}(s_t,a_t)[p^\diamond(s_t) - p(s_t;\xi_k)]+$$
$$Q_c^{\pi_{\theta_k}}(s_t,a_t)[1-p^\diamond(s_t)]]\nabla_\omega\lambda_\omega|_{\omega=\omega_k}]$$

Now, just as in the $\theta$ update convergence, we can demonstrate the following lemmas:

*Lemma 4*: $\delta\omega_{k+1}$ is square integrable since

$$\mathbb{E}[||\delta\omega_{k+1}||^2|\mathcal{F}_{\omega,k}] \leq 2\cdot\frac{H_{\max}}{1-\gamma}\cdot 1\cdot K_3(1+||\omega_k||_\infty^2) < \infty$$

for some large Lipschitz constant $K_3$. Note that $\mathcal{F}_{\omega,k} = \sigma(\omega_m, \delta\omega_m, m \le k)$ is the filtration for $\omega_k$ generated by different independent trajectories [8].

*Lemma 5*: Because $||Q_c(s_t, a_t; \kappa_k) - Q_c^{\pi_{\theta_k}}(s_t, a_t)||_\infty \to 0$ and $||p^\diamond(s_t) - p(s_t; \xi_k)||_\infty \to 0$ and $Q_c^{\pi_{\theta_k}}(s_t, a_t)[1 - p_{\xi^*}(s_t)]\nabla_\omega \lambda_\omega|_{\omega=\omega_k}$ is a sample of $Q_c^{\pi_{\theta^*}}(s_i, a_i)[1 - p_{\xi^*}(s_i)]\nabla_\omega \lambda_\omega|_{\omega=\omega_k}$, we conclude that $\mathbb{E}[\delta\omega_{k+1}|\mathcal{F}_{\omega,k}] = 0$ almost surely.

Thus, the lagrange multiplier $\omega$ update is a stochastic approximation of a continuous system $\omega(t)$ defined by [10]

$$\dot{\omega} = \Upsilon_\Omega[-\nabla_\omega L(\theta, \xi, \omega)|_{\theta=\theta^*(\omega),\xi=\xi^*(\omega)}] \tag{9}$$

with Martingale difference error of $\delta\omega_k$ and where $\Upsilon_\Omega$ is the left direction derivative defined similar to that in Time scale 2 of the convergence of $\theta$ update. Using Step 2 in Appendix A.2 from [8], we have that $dL(\theta^*(\omega), \xi^*(\omega), \omega)/dt = \nabla_\omega L(\theta, \xi, \omega)|_{\theta=\theta^*(\omega),\xi=\xi^*(\omega)}^T \cdot \Upsilon_\Omega[\nabla_\Omega L(\theta, \xi, \omega)|_{\theta=\theta^*(\omega),\xi=\xi^*(\omega)}] \ge 0$ and the value is non-zero if $||\Upsilon_\Omega[\nabla_\omega L(\theta, \xi, \omega)|_{\theta=\theta^*(\omega),\xi=\xi^*(\omega)}]|| \ne 0$.

For a local maximum point $\omega^*$, define a Lyapunov function as

$$\mathcal{L}(\omega) = L(\theta^*(\omega), \xi^*(\omega), \omega^*) - L(\theta^*(\omega), \xi^*(\omega), \omega)$$

Then there exists a ball centered at $\omega^*$ with a radius $\rho'$ such that $\forall \omega \in \mathfrak{B}_{\omega^*}(\rho') = \{\omega | ||\omega - \omega^*|| \le \rho'\}$, $\mathcal{L}(\omega)$ is a locally positive definite function, that is $\mathcal{L}(\omega) \ge 0$. Also, $d\mathcal{L}(\omega(t))/dt = -dL(\theta^*(\omega), \xi^*(\omega), \omega)/dt \le 0$ and is equal only when $\Upsilon_\Omega[\nabla_\omega L(\theta, \xi, \omega)|_{\theta=\theta^*(\omega),\xi=\xi^*(\omega)}] = 0$, so therefore $\omega^*$ is a stationary point. By leveraging Lyapunov theory for asymptotically stable systems presented in Chapter 4 of [55] we can demonstrate that for any initial conditions of $\omega \in \mathfrak{B}_{\omega^*}(\rho')$, the continuous state trajectory of $\omega(t)$ converges to the locally maximum point $\omega^*$.

Using these aforementioned properties, as well as the facts that 1) $\nabla_\omega L(\theta^*(\omega), \xi^*(\omega), \omega)$ is a Lipschitz function, 2) the step-sizes of Assumption on steps sizes, 3) $\{\omega_{k+1}\}$ is a stochastic approximation of $\omega(t)$ with a Martingale difference error, and 4) convex and compact properties in projections used, we can use Theorem 2 of chapter 6 in [10] to demonstrate the sequence $\{\omega_k\}$ converges almost surely to a locally maximum point $\omega^*$ almost surely, that is $L(\theta^*(\omega), \xi^*(\omega), \omega^*) \ge L(\theta^*(\omega), \xi^*(\omega), \omega)$.

From Time scales 2 and 3 we have that $L(\theta^*(\omega), \xi^*(\omega), \omega) \le L(\theta, \xi, \omega)$ while from Time scale 4 we have that $L(\theta^*(\omega), \xi^*(\omega), \omega^*) \ge L(\theta^*(\omega), \xi^*(\omega), \omega)$. Thus, $L(\theta^*(\omega), \xi^*(\omega), \omega) \le L(\theta^*(\omega), \xi^*(\omega), \omega^*) \le L(\theta, \xi, \omega^*)$. Therefore, $(\theta^*, \xi^*, \omega^*)$ is a local saddle point of $(\theta, \xi, \omega)$. Invoking the saddle point theorem of Proposition 5.1.6 in [54], we can conclude that $\pi(\cdot|\cdot; \theta^*)$ is a locally optimal policy for our proposed optimization formulation. $\square$

### C.4.4 Remark on Bounding Lagrange Multiplier

We can say our algorithm learns an REF that is closer to an optimally safe policy's REF as we take $\lambda_{\max} \to \infty$. Nonetheless, we want to put a bound on the $\lambda_{\max}$. This $\lambda_{\max}$ must be large enough so that choosing a policy that can reduce the expected cost returns by some non-zero amount is prioritized over increasing the reward returns. So any change in the reward critic terms must be less than any change in the cost critic term. If $H_\Delta$ is the minimum non-zero difference between any two cost values, and $P_{\min}$ is the minimum sampled non-zero likelihood of reaching a particular state and a point in the sample trajectory, then we can bound the maximum change in the reward returns and the maximum change on the weighted cost returns:

$$\Delta \mathop{\mathbb{E}}_{s \sim d_0}[V(s)] \le \frac{R_{\max}}{1-\gamma}$$

$$\gamma^T \cdot H_\Delta \cdot P_{\min} \cdot (\lambda + \frac{\phi(s)}{(1-\phi(s))}) \le \Delta \mathop{\mathbb{E}}_{s \sim d_0}[V_c(s) \cdot (\lambda + \frac{\phi(s)}{(1-\phi(s))})]$$

So we can find the bound for $\lambda_{\max}$:

$$\Delta \underset{s \sim d_0}{\mathbb{E}} [V(s) \cdot (1 - \phi(s))] < \Delta \underset{s \sim d_0}{\mathbb{E}} [V_c(s) \cdot (\lambda \cdot (1 - \phi(s)) + \phi(s))]$$

$$\Delta \underset{s \sim d_0}{\mathbb{E}} [V(s)] < \Delta \underset{s \sim d_0}{\mathbb{E}} [V_c(s) \cdot (\lambda + \frac{\phi(s)}{(1 - \phi(s))})]$$

$$\frac{R_{\max}}{1 - \gamma} < \gamma^T \cdot H_\Delta \cdot P_{\min} \cdot (\lambda + \frac{\phi(s)}{(1 - \phi(s))})$$

$$\frac{R_{\max}}{(1 - \gamma) \cdot \gamma^T \cdot H_\Delta \cdot P_{\min}} < \lambda + \frac{\phi(s)}{(1 - \phi(s))}$$

$$\frac{R_{\max}}{(1 - \gamma) \cdot \gamma^T \cdot H_\Delta \cdot P_{\min}} - \frac{\phi(s)}{(1 - \phi(s))} < \lambda$$

The second line holds since we are simply rearranging the comparative weightages of the reward and cost returns. Now $-\frac{\phi(s)}{(1-\phi(s))} \leq 0$ (recall that if $\phi(s) = 1$, then $\lambda$ is irrelevant in the lagrangian optimization). Thus, if $\lambda > \frac{R_{\max}}{(1-\gamma) \cdot \gamma^T \cdot H_\Delta \cdot P_{\min}}$ then minimizing the cost returns is prioritized over maximizing reward returns.

## D  Complete Experiment Details and Analysis

### D.1  Baselines

We compare our algorithm **RESPO** with 7 other safety RL baselines, which can be divided to CMDP class and hard constraints class, and unconstrained Vanilla PPO for reference.

CMDP Approaches

*Proximal Policy Optimization-Lagrangian.* **PPOLag** is a primal-dual method using Proximal Policy Optimization [33] based off of the implementation found in [7]. The lagrange multiplier is a scalar learnable parameter.

*Constraint-Rectified Policy Optimization.* **CRPO** [35] is a primal approach that switches between optimizing for rewards and minimizing constraint violations depending on whether the constraints are violated.

*Penalized Proximal Policy Optimization.* **P3O** [32] is another primal approach based on applying the technique of clipping the surrogate objectives found in PPO [33] to CMDPs.

*Projection-Based Constrained Policy Optimization.* **PCPO** [3] is a trust-region approach that takes a step in policy parameter space toward optimizing for reward and then projects this policy to the constraint set satisfying the CMDP expected cost constraints.

Hard Constraints Approaches

*Reachability Constrained Reinforcement Learning.* **RCRL** [27] is a primal-dual approach where the constraint is on the reachability value function and the lagrange multiplier is represented by a neural network parameterized by state.

*Control Barrier Function.* This **CBF**-based approach is inspired by the various energy-based certification approaches [56, 17, 18, 11, 57, 58]. This is implemented as a primal-dual approach where the control barrier-based constraint $\dot{h}(s) + \nu \cdot h(s) \leq 0$ is to ensure stabilization toward the safe set.

*Feasible Actor Critic.* **FAC** [9] is another primal-dual approach similar to **RCRL** (i.e. it uses the NN representation of the lagrange multiplier parameterized by state) except that the constraint in **FAC** is based on the cumulative discount sum of costs in lieu of the reachability value function. It is important to note that **FAC** is originally meant for the CMDP framework (with some positive cost threshold), but we adapt it to hard constraints by making the cost threshold $\chi = 0$. We do this to make a better comparison between an algorithm that relies on using the lagrange multiplier represented as a NN to learn feasibility with our approach of using our proposed REF function to learn the feasibility likelihood – both approaches enforce a hard constraint on the cumulative discounted costs.

## D.2 Benchmarks

We compare the algorithms on a diverse suite of environments including those from the Safety Gym, PyBullet, and MuJoCo suites and a multi-constraint, multi-drone environment.

*Safety Gym.* In Safety Gym [30], we examine CarGoal and PointButton which have 72D and 76D observation spaces that include lidar, accelerometer, gyro, magnetometer, velocimeter, joint angles, and joint velocities sensors. In the CarGoal environment, the car agent has a 72D observation space and is supposed to reach a goal region while avoiding both hazardous spaces and contact with fragile objects. In PointButton, the point agent has a 76D observation space and must press a series of specified goal buttons while avoiding 1) quickly moving objects, 2) hazardous spaces, 3) hitting the wrong buttons.

*Safety PyBullet.* In Safety PyBullet [50], we evaluate in BallRun and DroneCircle environments. In the BallRun environment, the ball agent must move as fast as possible under the constraint of a speed limit, and it must be within some boundaries. In DroneCircle, the agent is based on the AscTec Hummingbird quadrotor and is rewarded for moving clockwise in a circle of a fixed radius with the constraint of remaining within a safety zone demarcated by two boundaries. Note that we use this environment to evaluate our algorithm and the baselines in a stochastic setting. We ensure the MDP is stochastic by adding a $5\%$ gaussian noise to the transitions per step.

*Safety MuJoCo.* Furthermore, we compare the algorithms in with complex dynamics in MuJoCo. Specifically, we look at HalfCheetah and Reacher safety problems. In Safety HalfCheetah, the agent must move as quickly as possible in the forward direction without moving left of $x = -3$. However, unlike the standard HalfCheetah environment, the reward is based on the absolute value of the distance traveled. In this paradigm, it is easier for the agent to learn to quickly run backward rather than forward without any directional constraints. In the Safety Reacher environment, the robotic arm must reach a certain point while avoiding an unsafe region.

*Multi-Drone environment.* We also compare in an environment with multiple hard and soft constraints. The environment requires controlling two drones to pass through a tunnel one at a time while respecting certain distance requirements. The reward is given for quickly reaching the goal positions. The two hard constraints involve **(H1)** ensuring neither drone collides into the wall and **(H2)** the distance between the two drones is more than 0.5 to ensure they do not collide. The soft constraint is that the two drones are within 0.8 of each other to ensure real-world communication. It is preferable to prioritize hard constraint **H1** over hard constraint **H2**, since colliding with the wall may have more serious consequences to the drones rather than violations of an overly precautious distance constraint – as we will show, our algorithm **RESPO** can perform this prioritization in its optimization.

## D.3 Hyperparameters/Other Details

| Hyperparameters for Safe RL Algorithms | Values |
|:---:|:---:|
| **On-policy parameters** | |
| Network Architecture | MLP |
| Units per Hidden Layer | 256 |
| Numbers of Hidden Layers | 2 |
| Hidden Layer Activation Function | tanh |
| Actor/Critic Output Layer Activation Function | linear |
| Lagrange multiplier Output Layer Activation Function | softplus |
| Optimizer | Adam |
| Discount factor $\gamma$ | 0.99 |
| GAE lambda parameter | 0.97 |
| Clip Ratio | 0.2 |
| Target KL divergence | 0.1 |
| Total Env Interactions | $9e6$ |
| Reward/Cost Critic Learning rate | Linear Decay $1e-3 \rightarrow 0$ |
| Actor Learning rate | Linear Decay $3e-4 \rightarrow 0$ |
| Lagrange Multiplier Learning rate | Linear Decay $5e-5 \rightarrow 0$ |
| Number Seeds per algorithm per experiment | 5 |
| **RESPO specific parameters** | |
| REF Output Layer Activation Function | sigmoid |
| REF Learning rate | $1e-4 \rightarrow 0$ |
| **CBF specific parameters** | |
| $\nu$ | 0.2 |
| **RCRL/FAC Note** | |
| Lagrange Multiplier | 2-Layer, MLP (other algs just use scalar parameter) |

Table 2: Hyperparameter Settings Details

To ensure a fair comparison, the primal-dual based approaches and unconstrained Vanilla PPO were implemented based off of the same code base [59]. The other three approaches were implemented based on [60] with the similar corresponding hyperparameters as the primal-dual approaches. We run our experiments on Intel(R) Core(TM) i7-8700 CPU @ 3.20GHz with 6 cores. For Safety Gym, PyBullet, MuJoCo, and the multi-drone environments, each algorithm, per seed, per environment, takes $\sim 4$ hours to train.

## D.4 Double Integrator

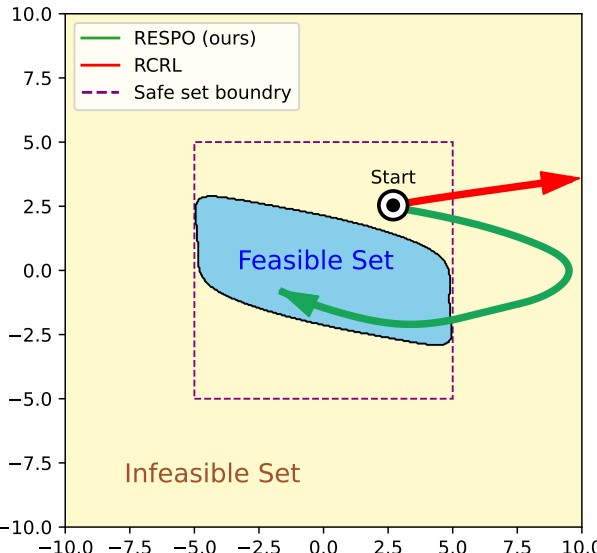

Figure 8: Comparison of the trajectories in the Double Integrator Environment of an agent controlled by policies obtained by RCRL (in red) and our proposed algorithm RESPO (in green) when starting from the infeasible set (but still within the safe set). Notice how our approach actively enters the feasible set (blue region), while RCRL fails to do so. The level set demarcating the feasible/infeasible set boundary is in black. The safe set (i.e. the set of states that have no violations) is the region within the dashed purple square. The infeasible set is in yellow.

We use the Double Integrator environment as a motivating example to demonstrate how performing constrained optimization using solely reachability-based value functions as in **RCRL** can produce nonoptimal behavior when the agent is outside the feasiblity set. Double Integrator has a 2 dimensional observation space $[x_1, x_2]$, 1 dimension action space $a \in [-0.5, 0.5]$, system dynamics is $\dot{s} = [x_2, a]$, and constraint as $||s||_\infty \leq 5$. Particularly, we make the cost as 1 if $||s||_\infty > 5$, and 0 otherwise to emphasize the importance of capturing the frequency of violation during training.

We train an **RCRL** controller and **RESPO** controller in this environment, and the results are visualized in Figure 8. The color scheme indicates the learned reachability value across the state space while the black line demarcates the border of the zero level set. We present the behavior of the trajectories of **RCRL** and **RESPO**. Because the **RCRL** optimizes for reachability value function when outside the feasible set, it simply minimizes the maximum violation, which as can be seen does not result in the agent reentering the feasible set since it is uniformly equal to or near 1 in the infeasible set. This is since it permits many violations of magnitude same or less than that of the maximum violation. On the other hand, **RESPO** optimizes for cumulative damage by considering total sum of costs, thereby re-entering the feasible set.

## D.5  Safety Gym Environments

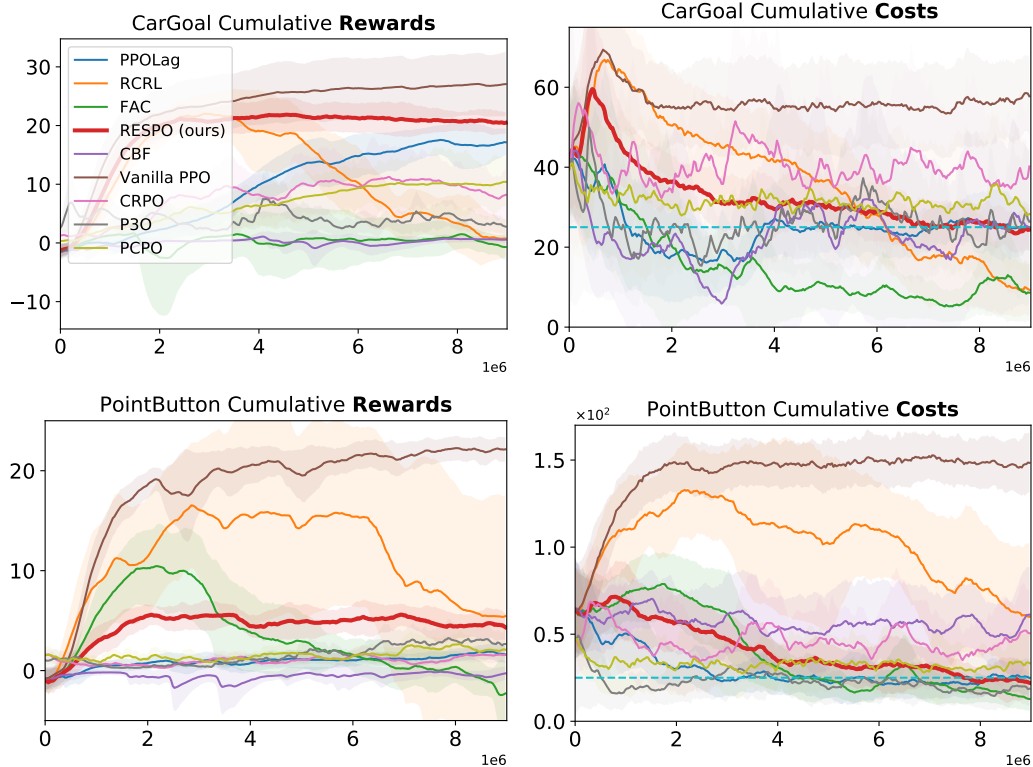

Figure 9: Closer look at comparison of algorithms in Safety Gym CarGoal and PointButton Environments.

**RESPO**: In the CarGoal Environment, our approach achieves the best performance among the safe RL algorithms while being within the acceptable range of cost violations. It is important to note that our algorithm *has no access to information* on the cost threshold. In PointButton, **RESPO** achieves very high reward performance among the safety baselines while maintaining among the lowest violations.

**PPOLag**: In both Safety Gym environments, **PPOLag** maintains relatively high performance, albeit less than our approach. Nonetheless, it always converges to the cost threshold amount of violations for the respective environments.

**RCRL**: **RCRL** has either high reward and high violations or low reward and low violations. It learns a very conservative behavior in CarGoal environment where the violations go down but the reward performance can also be seen to be sacrificed during training. For PointButton, **RCRL** achieves slightly higher reward performance but has over $3\times$ the number of violations as **RESPO**.

**FAC**: Using a NN to represent the lagrange multiplier in order to capture the feasible sets seems to produce very conservative behavior that sacrifices performance. In both CarGoal and PointButton the reward performance and cost violations are very low. This can be explained because the average observed lagrange multiplier across the states quickly grows, even becoming $9\times$ that of scalar learnable lagrange multiplier in **RESPO**.

**CBF**: The **CBF** approach has low reward performance in both the Safety Gym benchmarks and its cost violations are quite high.

**CRPO**, **P3O**, & **PCPO**: These CMDP-based primal approaches have mediocre reward performance but, with the exception of CRPO, achieve violations within the cost threshold. CRPO, however, has high cost violations in both CarGoal and PointButton.

**Vanilla PPO:** This unconstrained algorithm consistently has high rewards and high costs, so maximizing rewards does not improve costs in these environments.

## D.6 Safety PyBullet Environments

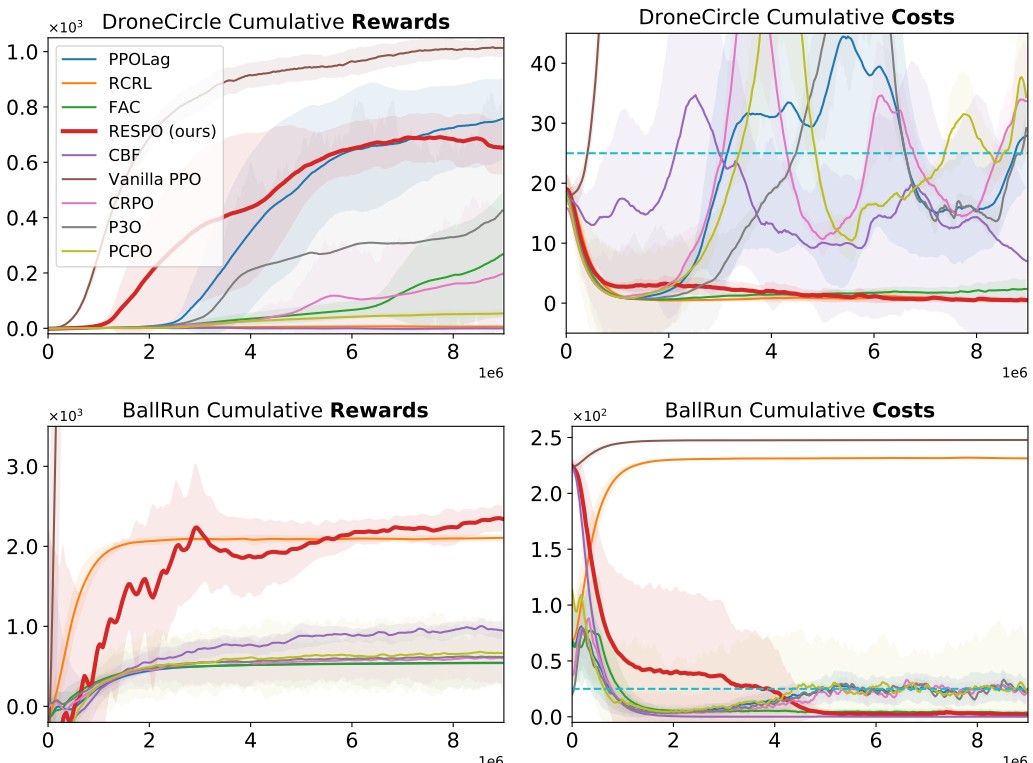

Figure 10: Closer look at comparison of algorithms in Safety PyBullet DroneCircle and BallRun Environments.

**RESPO**: In the both BallRun and DroneCircle, our approach achieves the highest or among the highest reward performance compared to the other safety baselines. Furthermore, **RESPO** converges to almost 0 constraint violations for both environments.

**PPOLag**: In DroneCircle, **PPOLag** has the best reward performance. Although its costs violations are around the cost threshold, it is much higher than **RESPO**. On the other hand, in BallRun, PPOLag has very low reward performance.

**RCRL**: We again see **RCRL** take on behavior with extremes – it has low reward and cost violations in Drone Circle and has high reward and cost violations in BallRun. Constraining the maximum violation with the reachability value function as **RCRL** does seems to provide poor safety in an environment with non-tangible constraints (i.e. a speed limit in BallRun).

**FAC**: While in BallRun, we see **FAC** have the same low reward and low violations behavior, DroneCircle shows an instance where **FAC** can achieve decently high rewards while maintaining low violations.

**CBF**: In DroneCircle, the **CBF** approach has low rewards and relatively low violations; in BallRun, it has a bit higher rewards compared to all the low performance algorithms with very low violations.

**CRPO**, **P3O**, & **PCPO**: These CMDP-based primal approaches have mediocre reward performance in DroneCircle and very low performance in BallRun. Nonetheless they achieve violations within the cost threshold.

**Vanilla PPO:** This unconstrained algorithm consistently has high rewards and high costs (sometimes out of the scope of the plots), so maximizing rewards does not improve costs in these environments.

## D.7 Safety MuJoCo Environments

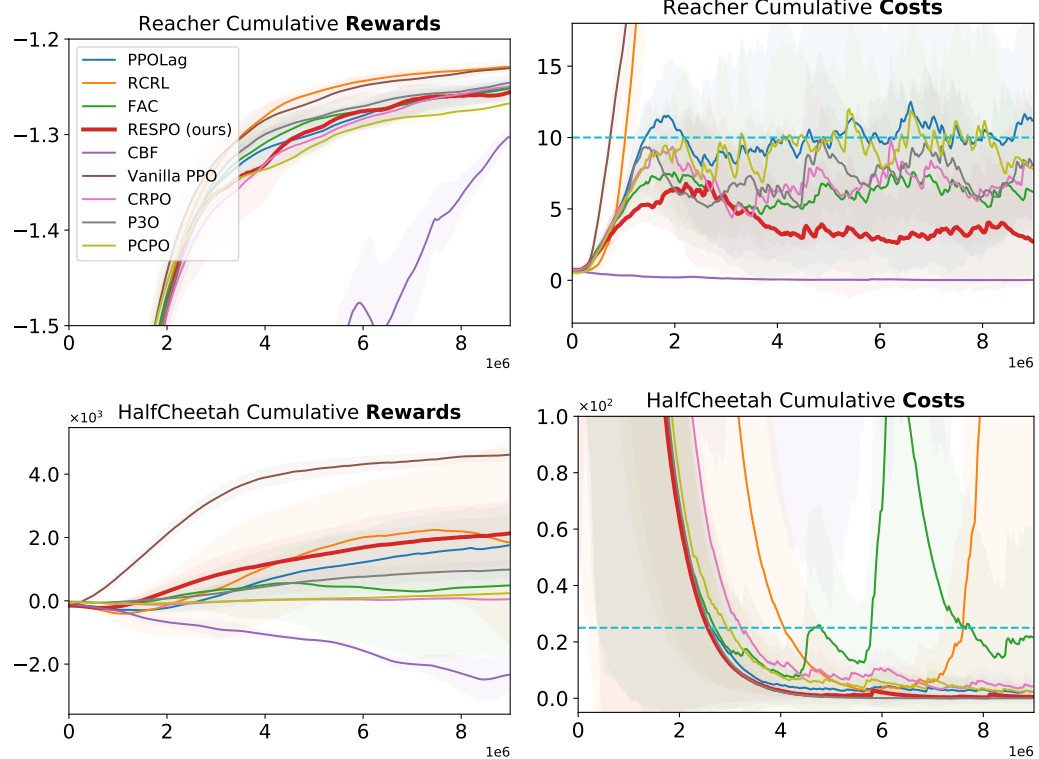

Figure 11: Closer look at comparison of algorithms in Safety MuJoCo Reacher and HalfCheetah Environments.

**Note on HalfCheetah**: The rewards for HalfCheetah are based on the absolute distance traveled in each step. Without the cost metric to constrain backward travel, it is easy to learn to run backward, as is the behavior learned in unconstrained PPO.

**RESPO**: Our approach achieves the highest reward performance among the safety baselines in HalfCheetah and decent reward performance in Reacher. Interestingly, **RESPO** also has 0 constraint violations in HalfCheetah and the second lowest constraint violations in Reacher.

**PPOLag**: The performance in Reacher for **PPOLag** is similar as in most of the previous environments: decently high rewrad, cost near the threshold. However, for HalfCheetah, interesting **PPOLag** learns to maintain the violations well below the cost threshold.

**RCRL**: We see yet again **RCRL** has high reward follow by very high constraint violations.

**FAC**: This approach has decent reward performance in Reacher and low reward performance in HalfCheetah. However, interestingly, **FAC** has high cost violations though below the cost threshold.

**CBF**: In Reacher, the **CBF** approach has conservative behavior with both low reward and low cost violations. But in HalfCheetah, it has very low reward performance and very high cost violations (not seen in the plot since its an order of magnitude larger than the visible range).

**CRPO**, **P3O**, & **PCPO**: These CMDP-based primal approaches have decent reward performance in Reacher while maintaining violations within cost threshold. In HalfCheetah, however, they achieve low performance and low cost violations.

**Vanilla PPO:** This unconstrained algorithm consistently has high rewards and high costs (sometimes out of the scope of the plots), so maximizing rewards does not improve costs in these environments.

## D.8 Hard and Soft Constraints

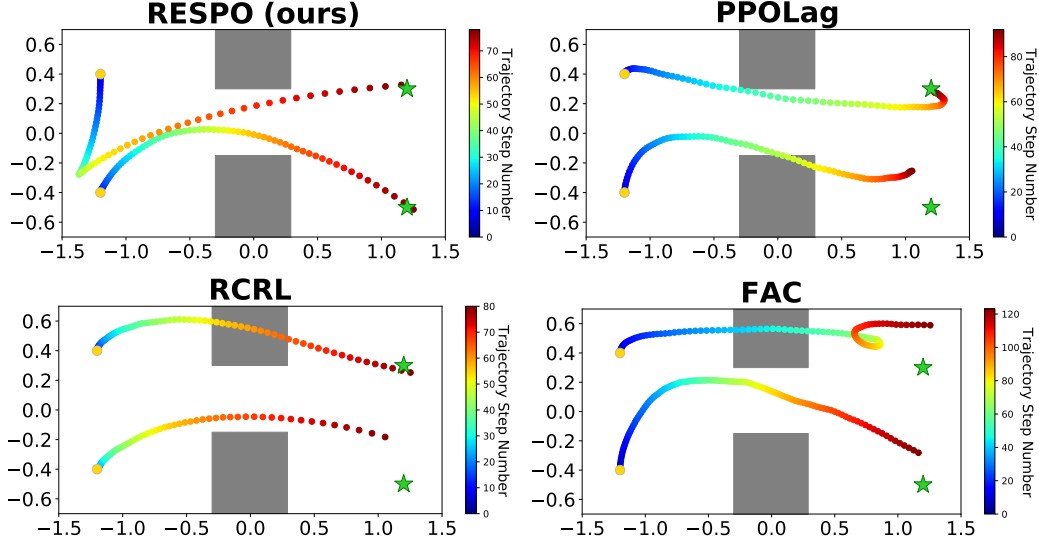

Figure 12: Closer look at comparison of RESPO with baselines trajectories in hard & soft Constraints multi-Drone control. Starting at gold circles, drones must enter the tunnel one at a time and reach green star while avoiding the wall and satisfying distance constraints. The colors indicate time along the trajectories.

**RESPO**: We manage multiple hard and soft Constraints by extending our framework to optimize the following Lagrangian:

$$
\min_{\pi} \max_{\lambda} \left( L(\pi, \lambda) = \mathbb{E}_{s \sim d_0} \left[ \left[ -V^{\pi}(s) + \lambda_{sc} \cdot (V_{sc}^{\pi}(s) - \chi) + \lambda_{hc2} \cdot V_{hc2}^{\pi}(s) \right] \cdot (1 - p_{hc2}(s)) \right. \right.
$$

$$
\left. \left. + V_{hc2}^{\pi}(s) \cdot p_{hc2}(s) + \lambda_{hc1} \cdot V_{hc1}^{\pi}(s) \right] \cdot (1 - p_{hc1}(s)) + V_{hc1}^{\pi}(s) \cdot p_{hc1}(s) \right] \right)
$$

(10)

The subscripts $hc1$ indicates the first hard constraint (i.e. wall avoidance), $hc2$ indicates the second hard constraint (i.e. drone cannot be too close), and $sc$ indicates soft constraint (i.e. drone cannot be too far) – they are all based on discounted sum of costs. Recall $V^{\pi}(s)$ is reward returns. We color coded the corresponding parts of the optimization. Notice how we learn a different REF for each hard constraint. Also notice that the feasible set of the first hard constraint $p_{hc1}$ is placed in a manner so as to ensure prioritization of the first hard constraint. As can be seen in the top left plot of Figure 12, our approach successfully reaches the goals and avoids the walls. To enable mobility of the top drone to pass through the tunnel with wall collision, the second hard constraint is violated temporarily in the blue to cyan time period. Furthermore to allow the bottom drone to pass through the tunnel, the soft constraint is violated during the green to orange time period. Nonetheless, **RESPO** successfully manages the constraints and reward performance via Equation 10 optimization.

**PPOLag, RCRL, FAC**: The optimization formulation for these approaches is as follows:

$$
\min_{\pi} \max_{\lambda} \left( L(\pi, \lambda) = -V^{\pi}(s) + \lambda_{sc} \cdot (V_{sc}^{\pi}(s) - \chi) + \lambda_{hc2} \cdot V_{hc2}^{\pi}(s) + \lambda_{hc1} \cdot V_{hc1}^{\pi}(s) \right) \quad (11)
$$

For **PPOLag** and **FAC**, all the constraint value functions are discount sum of cost. For **RCRL**, $V_{sc}^{\pi}(s)$ is based on discount sum of costs but $V_{hc1}^{\pi}(s)$ and $V_{hc2}^{\pi}(s)$ are based on the reachability value function. Furthermore, in **PPOLag**, all the lagrange multipliers are learnable scalar parameters. In **FAC** and **RCRL** all the lagrange multipliers are NN representations parameterized by state. These formulations are not able to provide a framework for the prioritization of the constraint satisfaction – all the constraints are treated the same, weighted only on the learned lagrange multipliers. As can be seen in the other three images in Figure 12, the algorithms cannot manage the multiple constraints, and invariably collide with the wall.

## D.9    Ablation – Learning Rate

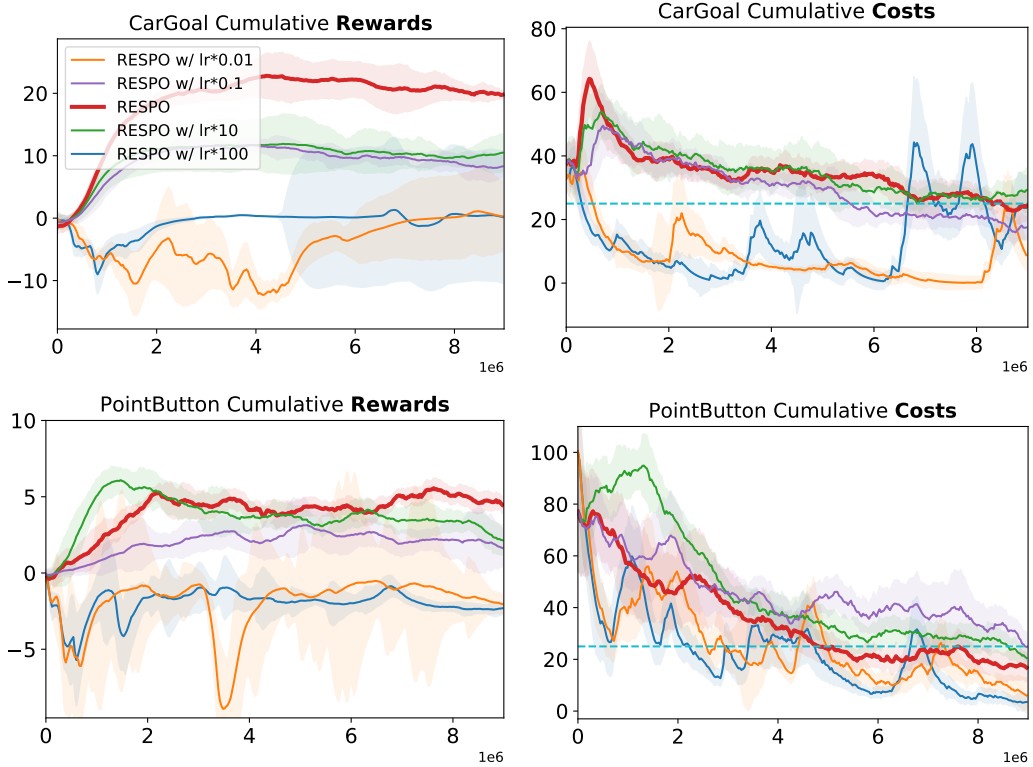

Figure 13: Closer look at Ablation study on the learning rate of REF.

We performance Ablation study of varying the learning rate of the REF function to verify the importance of the multi-timescale assumption. Particular, we compare our algorithm's approach of placing the learning rate of the REF between the policy and lagrange multiplier with making the REF's learning rate in various orders of magnitudes slower and faster. Our approach with the learning rate satisfying the multi-timescale assumption experimentally appears to still have the best balance of reward optimization and constraint satisfaction. Particularly when we change the learning rate by one order of magnitude (i.e. $\times 10$ or $\times 0.1$), we see the reward performance reduce by around half and while the cost violations generally don't change. But when we change the learning rates by another order of magnitude, there reward performance effective becomes zero and the cost violations generally reduce further. By increasing the learning rate of the REF function, we can no longer guarantee that the REF convergences to near the optimally safe REF value. Instead, it becomes the REF of the policy in question. So instead, the optimization can learn to "hack" the REF function to obtain a policy (and lagrange multiplier) that is not a local optimal for the optimization formulation. On the other hand, when the learning rate is too slow, the lagrange multiplier quickly explodes, thereby creating a very conservative solution – notice the similarity of the orange line in Figure 13 with learning rate 0.01 times that of standard in training behavior with **PPOLag** where $\chi = 0$ in the ablation study on optimization.

## D.10 Ablation – Optimization

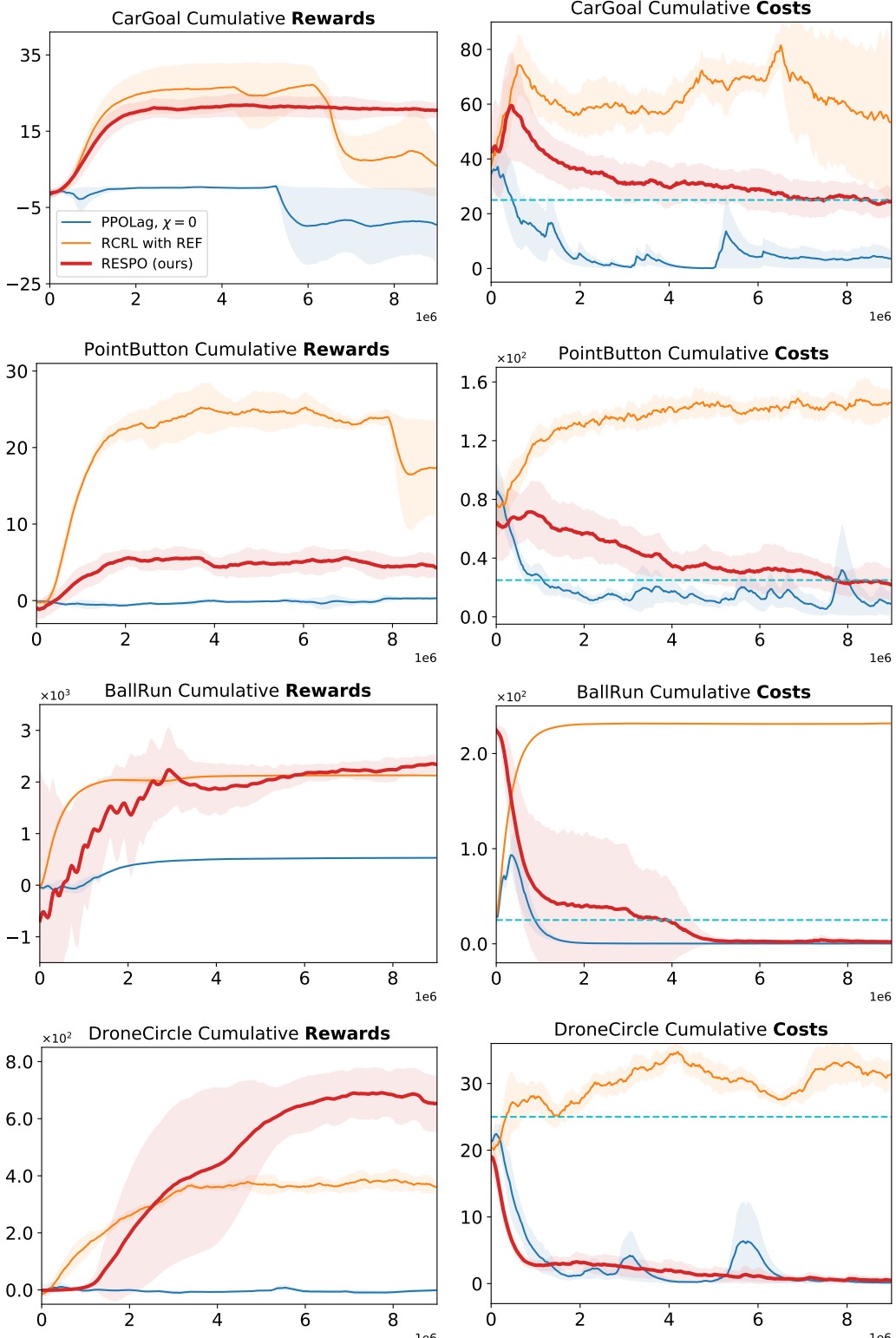

Figure 14: Closer look at Ablation study on hard constraints optimization frameworks.

In this ablation study, we examine the various optimization frameworks within the context of hard constraints. Particularly, we compare our **RESPO** framework with **RCRL** and **CMDP**. However, for **RCRL** we implement using our REF function method while still keeping the reachability value function. For **CMDP**, we make the cost threshold $\chi = 0$. These comparisons answer important questions about our design choices – specifically is it sufficient to simply to just use the REF component or to just learn the cost returns alone? From this ablation study, we propose that though we have provided theoretical support for adding each of these design components individually, in practice they are both required together in our algorithm. In **RCRL** implemented with our REF, we generally see decently high rewards but the cost violations are always very large. This highlights the problems of the reachability function again – if the agent starts or ever wanders into the infeasible set, there is no guarantee of (re)entrance into the feasible set. So the agent can indefinitely remain in the infeasible set, thereby incurring potentially an unlimited number of constraint violations. In **PPOLag** with $\chi = 0$, both the reward performance and constraint violations are very low. By using such hard constraints versions of these purely learning-based methods, even when using the cumulative discounted cost rather than reachability value function, the reward performance is very low because the lagrange multiplier becomes too large quickly and thereby overshadows the reward returns in the optimization. Ultimately, both the REF approach and the usage of the cumulative discounted costs are important components of our algorithm **RESPO** that encourage a good balance between the reward performance and safety constraint satisfaction in such stochastic settings.

