# OpenReview forum: "Iterative Reachability Estimation for Safe Reinforcement Learning"
_NeurIPS.cc/2023/Conference — NeurIPS 2023 poster_

### Official Review · Reviewer_hXDB · 2023-06-16

**Soundness:** 4 excellent
**Presentation:** 3 good
**Contribution:** 3 good
**Rating:** 8
**Confidence:** 5

**Summary:**

This paper proposed an iterative reachability estimation method for safe RL. The reachability is estimated by the probability of future trajectories entering unsafe state sets. Compare to previous reachability-based methods, the proposed method could handle stochastic dynamics and also improved the performance with deterministic dynamics. The proposed algorithm also leverages more information from data which could explain the performance improvement compared to existing methods. Theoretical convergence results are provided. Experimental results well supported the claimed performance improvement and safety guarantees.

This paper has removed one significant limitation, the deterministic dynamics assumption in previous studies. Removing this limitation should be very careful, and I think the authors did it very well. The author also connected HJ reachability with CMDP, which is two important definitions in the safe RL. Therefore, I strongly suggested the paper be accepted.

**Strengths:**

Originality:
Very good. As far as I now, no paper has considered stochastic reachability with the constrained RL setup. The authors also did a good job of connecting HJ reachability and CMDP, which are two important definitions in constrained RL that used to be separately considered. This combination also removed significant limitations, the deterministic dynamics of the previous study.

Quality:
Excellent. The paper very clearly explained the relation and improvement with respect to the previous paper in both deterministic and stochastic settings and did a comprehensive comparison in the experimental section. The authors also summarized the novelty and advantages of intuitions which is very easy to understand, i.e., previous methods only consider the maximum violation, which might lose information in the whole episode.

Clarity:
The presentation of this paper is good.

Significance:
The safety of stochastic systems is very important and challenging. There have been many theories and studies to formulate the problem, and reachability is undoubtedly one of the most powerful methods. Handling stochastic reachability should be very careful.

**Weaknesses:**

I feel good about most of the paper, I only have comments on some minor problems:
1. The problem formulation, equation (4) should be emphasized better so that the reader will know this is the proposed problem formulation.
2. The notation system is a bit messy. The readers might get lost easily, especially those not familiar with the previous paper.
3. Algorithm 1 actually did not provide too much useful information. You should improve it to highlight the differences between your algorithm and the previous ones, like the REF update.

**Questions:**

I like most of the intuitions to explain the advantages of this paper. However, I have some questions:

1. (line 179 - 181, about the claimed limitations of previous work RCRL) _**These improvements in costs can be crucial in guiding the optimization toward a safer policy.** And optimizing with $V_h(s) $ can result in accumulating an unlimited number of violations smaller than the maximum violation._

I roughly got the intuition. I think you mean that some policies might be too far away from the safe policy. The optimization landscape looks like $V_h(s)$ should be non-decreasing in the first few steps but finally decrease to a low value. This is indeed interesting. Because from my intuition, a good policy update should always point to the direction that $V_h(s)$ decreases.

Did you observe this in the experiment? If so, could you show me this phenomenon with some experimental results? It would be super helpful for enhancing the contribution of this paper.

2. (Section 5.3, about the REF and multiplier.) The REF update actually relies on the distribution density function. It seems like you did not include this paper in the references,
> Qin, Zengyi, Yuxiao Chen, and Chuchu Fan. "Density constrained reinforcement learning." International Conference on Machine Learning. PMLR, 2021.
I think it is a good paper to further understand the relationship between density function and Lagrange multipliers in safe RL problems. You should discuss the relationship with this paper.

**Limitations:**

The authors have adequately addressed the limitations.

---

> ### Author Rebuttal · Authors · 2023-08-07
>
> We thank the reviewer for the comments and the detailed suggestions.
>
> >The problem formulation, equation (4) should be emphasized better... The notation system is a bit messy ... You should improve it to highlight the differences between your algorithm and the previous ones
>
> Thank you, we will further emphasize the equation, clarify the notations, and include the full form of the gradient of the losses in the algorithm.
>
> > The optimization landscape looks like $V_h$ should be non-decreasing in the first few steps but finally decrease to a low value. … Did you observe this in the experiment? If so, could you show me this phenomenon with some experimental results?
>
> Yes, similar to what the reviewer suggests. In the quote, we are describing that since RCRL just computes maximum cost violation, any changes in cost later on in the trajectory that remain lower than the maximum cost will not affect the RCRL value. However, with cumulative sum of costs as used in RESPO, we can gain more learning signal since any change in costs along the trajectory will affect the cumulative sum.
>
> This phenomenon explains, in the case when the agent is outside the feasible set, why RCRL tends to remain outside the feasible set and the RCRL optimization may not point to the direction that $V_h$ strictly decreases. However, our approach, RESPO, tends to (re)enter the feasible set.
>
> For a visual on experimental results for comparing this property of cumulative costs compared to maximum reachable cost, we implemented a Double Integrator example and compared the trajectories between RESPO and RCRL in Appendix Section 4.4. It demonstrates how when the agent begins outside the feasible set, the _trajectory from RCRL remains outside the feasible set_ while the _trajectory from RESPO enters back into the feasible set_. For more information on this behavior of (re)entrance into the feasible set, we also refer the reviewer to Proposition 2 (line 213 of main paper) and its proof in Appendix Section 3.3.
>
> > The REF update actually relies on the distribution density function. It seems like you did not include this paper in the references
>
> Thank you, we will include it in the related work. Our REF estimates likelihood of future violations and is computed through reachability bellman formulation using the max operator. On the other hand, [Qin et al.]’s density metric computes state visitation density and is defined by the discounted sum of the likelihood of visiting a particular state. We will add more discussion on the relationship and differences between RESPO and [Qin et al.] in our related works.
>
> [Qin et al.] Density constrained reinforcement learning. In ICML. PMLR, 2021.

---

> > ### Comment · Reviewer_hXDB · 2023-08-18
> >
> > Thank you for the reply. I am now very confident that it is a good paper and the experimental results are convincing and comprehensive.

---

### Official Review · Reviewer_teXi · 2023-07-01

**Soundness:** 3 good
**Presentation:** 2 fair
**Contribution:** 3 good
**Rating:** 7
**Confidence:** 4

**Summary:**

This paper presents  Reachability Estimation for Safe Policy Optimization (RESPO), for safety-constrained RL in general stochastic settings. The authors extend the previous RCRL approach into stochastic settings and push the agent to (re)enter the feasible region. They formulate a safe RL problem with REF and further develop an adapted AC algorithm to solve it, with convergence analysis. They compare their approach against CMDP-based approaches with a soft constraint and RCRL with a hard constraint, showing the advantages.

**Strengths:**

The paper is well-motivated and well-structured to follow. It studies a critical problem.
Based on RCRL, this paper does have some novelties in problem formulation and proposed approach. Technically, this paper is sound to me, although I only checked part of the math proof in the appendix.
The experiments are promising as they show mixed performance and safety violation improvements.



**Weaknesses:**

1. The writing could be further improved, especially the comparison with RCRL. The reviewer acknowledges that there is some explanation of the difference between the proposed approach and RCRL, still, it would be much better to add more and clarify it. For example, the reviewer is confused why RCRL cannot guarantee or optimize (re)entrance to the feasible set. Couldn't use the same proof of Proposition 2 to obtain the same (re)entrance proposition?

2. For the deterministic environments, my understanding is RCRL considers a harder constraint as it is per state constraint than the discounted additive constraints in your paper, why the harder constraint cannot optimize/guarantee (re)entrance to the feasible set?

3. How does the reentrance proved from the deterministic environment applied to stochastic systems?

3. There are some recent safe RL papers considering hard constraints. For example,

 a. Wang, Y., Zhan, S. S., Jiao, R., Wang, Z., Jin, W., Yang, Z., ... & Zhu, Q. (2022). Enforcing hard constraints with soft barriers: Safe reinforcement learning in unknown stochastic environments. ICML 2023.

 b.  Xiong, N. (2023). Provably Safe Reinforcement Learning with Step-wise Violation Constraints. arXiv preprint arXiv:2302.06064.
The authors may consider talking about these recent references in the paper revision.

4. what do you mean by "almost surely" in the convergence analysis?

**Questions:**

Please see the weakness.

**Limitations:**

What are the limitations of RESPO?

---

> ### Author Rebuttal · Authors · 2023-08-07
>
> We thank the reviewer for the comments and detailed suggestions.
>
> > The writing could be further improved, especially the comparison with RCRL. The reviewer acknowledges that there is some explanation of the difference between the proposed approach and RCRL, still, it would be much better to add more and clarify it.
>
> We will make further explanations along these lines. In lines 170-185, we listed several specific problems in RCRL that we aim to address, namely: 1) RCRL is limited to the deterministic setting and 2) RCRL cannot guarantee (re)entrance into the feasible set. In particular, RCRL is limited to deterministic settings because its reachability value function in the Bellman formulation does not directly apply to the stochastic setting.
>
> >Why RCRL cannot guarantee or optimize (re)entrance to the feasible set. Couldn't use the same proof of Proposition 2 to obtain the same (re)entrance proposition?
>
> RCRL cannot guarantee (re)entrance because its reachability value function permits having many (possibly infinite) violations less than or equal to the maximum violation. For a small example, consider the agent faced with two possible trajectories: trajectory A which has a violation sequence (100, 0, 100) and trajectory B which has a violation sequence (100, 0, 0). The reachability value function would assign value 100 to both trajectory A and B since 100 is the maximum violation. So a reachability value approach like RCRL would choose both paths with equal likelihood and not be guaranteed to enter the feasible set. However, by considering the cumulative violations, trajectory A will have a greater cost score than B. Therefore, an approach using cumulative cost like RESPO will optimally choose trajectory B and thereby enter the feasible set.
>
> We proposed to utilize reachability to instead measure likelihood of violation and to use (discounted) sum of costs along a trajectory to minimize cumulative violations. As Proposition 2 indicates, the (discounted) sum of costs can guarantee (re)entrance into the feasible set whenever possible by not allowing an uncountable number of violations smaller than (or equal to) the maximum violation in the trajectory. For details on the proof, please refer to Appendix section 3.3. For experimental support for this proposition, please refer to the Double Integrator example in Appendix section 4.4 which considers the behavior under RCRL versus RESPO when the agent starts inside the safe set but outside the feasible set.
>
> >For the deterministic environments, my understanding is RCRL considers a harder constraint as it is per state constraint than the discounted additive constraints in your paper, why the harder constraint cannot optimize/guarantee (re)entrance to the feasible set?
>
> A hard, state-wise constraint approach doesn’t necessarily guarantee entrance into the feasible set when the agent begins _outside the feasible set_. RCRL is unable to (re)enter the feasible set because of the formulation of its reachability value function. Specifically, because it only minimizes the maximum violation along a trajectory, it does not consider the possibility of many (potentially infinite) violations smaller than or equal to the maximum that prevent it from entering the feasible set.
>
> > How does the reentrance proved from the deterministic environment applied to stochastic systems?
>
> The principle of reentrance using cumulative costs can be extended to stochastic systems. The basic idea we are providing in Proposition 2 is that if there exists a way to (re)enter the feasible set, this path would have a finite cumulative cost and all trajectories that do not enter the feasible set will have infinite cost (for gamma close to 1 and very large horizons), therefore the optimization will produce a policy with minimal (i.e. finite) cost which reenters the feasible set. This principle will also be reflected in stochastic systems (consider that even if the chance of ending up in a trajectory with infinite cost is small but nonzero, the expected cumulative cost will still be infinite), which justifies the usage of a value function based on cumulative cost rather than maximum cost along a trajectory.
>
> > There are some recent safe RL papers considering hard constraints. For example, [Wang et al.] and [Xiong et al.]
>
> Thank you, we will add these papers in our related works section. Both these papers are model-based while our proposed algorithm falls in the category of model-free safe RL.
>
> > what do you mean by "almost surely" in the convergence analysis?
>
> By almost surely, we are using the probability theory definition that the likelihood of the event (i.e. convergence to local optimum) is 1. This definition has been used in past works including [Chow et al.] and [Borkar].
>
> > What are the limitations of RESPO?
>
> We refer the reviewer to main paper lines 358-360. The main limitation is that we do not guarantee minimal violations _during_ training. We leave open potential extensions that can help in applications like safety in single-lifetime reinforcement learning.
>
>
> [Chow et al.]  Risk-constrained reinforcement learning with percentile risk criteria. JMLR, 2017.
>
> [Borkar] Stochastic approximation: a dynamical systems viewpoint, volume 48. Springer, 2009.

---

> > ### Comment · Reviewer_teXi · 2023-08-12
> >
> > Thanks for the detailed rebuttals. The responses have fully addressed my questions and I am willing to increase my score to a 7.

---

### Official Review · Reviewer_YH1f · 2023-07-03

**Soundness:** 3 good
**Presentation:** 3 good
**Contribution:** 3 good
**Rating:** 7
**Confidence:** 4

**Summary:**

Previous approaches to safe reinforcement learning used the constrained MDP formulation where there is a constraint imposed on the cumulative sum of costs to minimise violations. This framework is not applicable very easily where there is a need for hard constraint satisfaction. The previous approach (RCRL) which leverages reachability analysis to strictly satisfy hard constraints is limited to deterministic MDP and is not suited to bringing the state back to the feasible set when already outside the feasible set. In this work, the authors minimize the expected chance and frequency of violations under stochastic transition dynamics thus resolving the two problems with the previous work. The order in which to update the Q-networks, the policy networks, the Lagrangian dual factors and the reachability estimation function is studied using an empirical approach and a theoretical convergence guarantee to a local optimum is provided for this alternating optimization. Empirical comparisons are made extensively to a wide spectrum of existing approaches.

**Strengths:**

Advantages of paper in relation to deterministic dynamics assumption of RCRL is clear and has merit. Empirical comparisons to previous work are quite extensive with additional explanations in the appendix. Convergence to local optimum is presented to establish rigor and soundness of the method.


**Weaknesses:**

1) In section 6, it was not very clear what to see in the figures and it looked like RESPO was achieving a different point in the trade-off curve compared to the other methods. More information was available in the appendix. More discussion on how to compare the different methods and why one method performs better in a specific metric can be written out in the main section. The motivation of the paper provides the twin advantages of getting back into the feasible set and accounting for stochasticity. The first advantage is seen in the double integrator example in the appendix. Do any of the previous methods suffer due to deterministic assumptions and is the actual MDP stochastic?

2) Since there are many networks and parameters updated at the same time, the robustness and reproducibility of the training process for this method and similar previous methods seem suspect. The authors have discussed this aspect in the ablation studies and the best possible convergence is obtained in the way the authors are doing the training. This insight, though a mild weakness, could benefit the community as we are inferring new insights about alternating optimization between multiple networks tied to each other.


**Questions:**

1) While doing the comparison to CBF methods, it is not clearly what is being compared against. CBFs/Energy functions are usually handcrafted but there is CBVF (cited by the paper) and other related work that tries to construct the optimal CBF with maximum safety set (assuming known model). With the optimal CBF, the behaviour of CBF methods is not very conservative. If stochasticity is the source of trouble for the CBF methods, some extensions such as robust CBF and learned CBF are available. Here, the authors can provide more details about which exact CBF method they are using.

2) In line 155, should $V_c$ be in the expression for optimal policy? $V_c$ is previously defined differently. “We define optimal REF based on an optimally safe policy π∗” is then misleading. The policy is optimizing a different loss in line 227 and it is not clear whether we actually get the optimal REF from this process.

3) Assumption A3 (Lipschitz gradients) is a rather strong assumption. Are there previous cases where RL value functions are assumed to have Lipschitz gradients?

4) In figure 3, why are RCRL and FAC violating the hard constraint if they are designed to respect hard constraints?

5) The variance of the red RESPO learning curves seem to be high in certain figures indicating training instabilities


**Limitations:**

Authors have adequately addressed the limitations.

---

> ### Author Rebuttal · Authors · 2023-08-07
>
> We thank the reviewer for the comments and detailed suggestions.
>
> > it looked like RESPO was achieving a different point in the trade-off curve compared to the other methods.
>
> Our approach consistently achieves higher rewards and lower costs compared to the other safety baselines. Particularly, RESPO almost always achieves the highest or second highest reward performance among the safety baselines with low cumulative costs. When RESPO achieves the second highest, the algorithm with the highest (RCRL or PPOLag) always incurs several times more cost, generally beyond the acceptable threshold for that environment. The exception to this is the Reacher environment, but still RESPO has similar performance as other performance-successful algorithms and significantly lower cost than them. We will add more discussion on this in the main paper.
>
> >  Do any of the previous methods suffer due to deterministic assumptions and is the actual MDP stochastic?
>
> We ensured environments running with the PyBullet engine, namely DroneCircle and BallRun, were stochastic by adding some noise. We will explain more details on this in the paper. Note that the policies learned were stochastic (i.e. $\pi(a|s)$). We can see poor performance (i.e. either low rewards and low costs or high rewards and high costs) of algorithms like RCRL that assume deterministic environments.
>
> > This insight, though a mild weakness, could benefit the community as we are inferring new insights about alternating optimization between multiple networks tied to each other.
>
> As the reviewer noted earlier, we demonstrated via 1) ablation studies in figure 5 on varying learning rates as well as 2) the proof in Appendix section 3.4 that having a particular order of learning rate schedules for the networks satisfying Assumption A1 (main paper line 265) guarantees convergence to the local optimum of our RESPO optimization.
>
> > If stochasticity is the source of trouble for the CBF methods, some extensions such as robust CBF and learned CBF are available.
>
> As per the reviewer's suggestion, we have run additional experiments using model-free learned CBF certificate method found in [Yang et al] and results can be seen in pdf attached to global response. We see that compared to our approach and other baselines, the learned CBF produces conservative behavior (low reward and lost cost) likely because it is difficult to obtain the optimal CBF.
>
> > the authors can provide more details about which exact CBF method they are using.
>
> More details about the CBF method can be found in the Appendix lines 290-292. Particularly, the constraint is $(c’ - c)/dt + \nu \cdot c \leq 0$ where $c$ and $c’$ are consecutive cost values in a trajectory and $dt$ is the time step. This approach was also used as the CBF baseline in the RCRL paper.
>
> >In line 155, should $V\_c$ be in the expression for optimal policy? ... The policy is optimizing a different loss in line 227 and it is not clear whether we actually get the optimal REF from this process.
>
> Yes, $V_c$ should be in the expression. The RESPO optimization we proposed in line 227 will optimize for reward (i.e. $V^{\pi}$) while maintaining cost $V^{\pi}\_{c} =0$ if the state is in the feasible set and minimize the cost $V^{\pi}\_{c}$ if the state is in the infeasible set. Notice, this produces similar safety as $\pi^* = \arg \min\_{\pi} V^{\pi}\_c (s)$ : If $ V^{\pi} \_c (s)=0 $ then the state is in the feasible set and the RESPO optimization maximizes reward $V^{\pi}$ while maintaining the cost constraint of 0 violations. In the other case, $\min\_{\pi} V^{\pi}\_c(s)$ and RESPO behavior both minimize the cumulative cost. Since in both cases RESPO ensures $V^{\pi}\_c(s)$ is minimum, therefore the optimal policy of RESPO is actually an optimal policy of $\min\_{\pi} V^{\pi}\_c (s)$. And hence, we can define the optimal REF from RESPO “based on __an__ optimally safe policy $\pi^∗$.”
>
> > Are there previous cases where RL value functions are assumed to have Lipschitz gradients?
>
> This assumption has been made in the past in several papers including [Chow et al] and [Yu et al]. Note we only use linear, tanh, softplus, and sigmoid activation functions which have Lipschitz continuous gradients.
>
> > In figure 3, why are RCRL and FAC violating the hard constraint if they are designed to respect hard constraints?
>
> Approaches like FAC and RCRL are unable to converge to a suitable policy that manages the importance of the constraints because they treat the soft constraint and both the hard constraints all with the same priority. However, our approach can prioritize the constraints within its optimization formulation (more details on how we do it are in appendix lines 423-435). Particularly RESPO never violates the primary constraint of wall avoidance and it only rarely violates the secondary hard constraint of closeness of the drones. Please refer to Section 4.8 of the appendix for more details on the optimization formulation for RCRL and FAC (particularly line 436).
>
> >The variance of the red RESPO learning curves seem to be high in certain figures indicating training instabilities.
>
> RESPO actually demonstrates smaller instabilities compared to other baseline algorithms near the end of training. While in some environments there are instabilities in the beginning of training of RESPO, our experiments show that RESPO generally converges to a consensus later in training. The only environment where the variance is relatively high at the end of training is DroneCircle—but even still, the baseline approaches with decent reward performance in DroneCircle, like PPOLag, P3O, FAC, and CRPO, have an even higher variance than RESPO.
>
> [Yang et al.] Model-free safe reinforcement learning through Neural Barrier Certificate. IEEE Robotics and Automation Letters, 2023.
>
> [Chow et al.]  Risk-constrained reinforcement learning with percentile risk criteria. JMLR, 2017.
>
> [Yu et al.] Reachability constrained reinforcement learning. In ICML. PMLR, 2022.

---

> > ### Comment · Reviewer_YH1f · 2023-08-12
> > **Good job**
> >
> > I believe this is a strong paper. The authors have addressed all questions and provided clarifications. I have read the other reviews and comments. Overall, I am happy to increase my score to 7 at this time. I will stay tuned to see whether there are any further questions from other reviewers.

---

### Official Review · Reviewer_yyXc · 2023-07-05

**Soundness:** 3 good
**Presentation:** 2 fair
**Contribution:** 2 fair
**Rating:** 5
**Confidence:** 3

**Summary:**

The paper proposes a new algorithm that may handle hard and soft constraints, in which the policy optimization and Hamilton-Jacobi reachability are leveraged to ensure safety. Moreover,  experiment results on safety gym, safety PyBullet, and safety MuJoCo also show the good performance of their algorithm.

**Strengths:**

1. The convergency analysis sounds good.
2. Comprehensive experiments are provided.
3. Hard constraints and soft constraints are investigated.

**Weaknesses:**

1. Paper writing quality needs to be improved a lot, I am confused about the paper notation, e.g., V_h and V_c.
2. The experimental results are not correct regarding some baselines, especially for CRPO, in CRPO paper, the algorithm presents better performance than PPO-Lagrangian and CPO.

**Questions:**

1. Could you analyze the difference between RESPO and CRPO regarding the reward update and cost update?
2. If the agent does not find the safe action, will it be stuck at a point?
3. How do you define the reachable set when considering reward performance?

**Limitations:**

1. The balance between reward and cost is not addressed well, in the experiments, as shown in Figure 3, although RESPO can ensure safety, the trajectory is longer than other baselines, and also not smooth.
2. Some related papers are not mentioned in the study, e.g., [Kochdumper, N., et al., 2023], [Gu, S., et al., 2022] and [Selim, M., et al., 2022].

[Kochdumper, N., et al., 2023] Kochdumper, N., Krasowski, H., Wang, X., Bak, S., & Althoff, M. (2023). Provably safe reinforcement learning via action projection using reachability analysis and polynomial zonotopes. IEEE Open Journal of Control Systems, 2, 79-92.

[Gu, S., et al., 2022]  Gu, S., Chen, G., Zhang, L., Hou, J., Hu, Y., & Knoll, A. (2022). Constrained reinforcement learning for vehicle motion planning with topological reachability analysis. Robotics, 11(4), 81.

[Selim, M., et al., 2022] Selim, M., Alanwar, A., Kousik, S., Gao, G., Pavone, M., & Johansson, K. H. (2022). Safe reinforcement learning using black-box reachability analysis. IEEE Robotics and Automation Letters, 7(4), 10665-10672.

---

> ### Author Rebuttal · Authors · 2023-08-07
>
> We thank the reviewer for the comments and detailed suggestions.
>
> >Paper writing quality needs to be improved a lot, I am confused about the paper notation, e.g., $V\_h$ and $V\_c$.
>
> We have provided the definitions of the notations in lines 108-165 of the main paper, as well as a summary of notation in Table 1 of the appendix. We will further revise the notation to be more clear. $V_h$ is the reachability value function defined in Definition 2 (line 145 of main paper) which describes the maximum reachable violation. $V_c$ is the cumulative discounted sum defined in the CMDP section (line 125 of main paper).
>
> >The experimental results are not correct regarding some baselines, especially for CRPO, in CRPO paper, the algorithm presents better performance than PPO-Lagrangian and CPO.
>
> With due respect, we don't believe reviewer's claim is accurate. The experiments in the CRPO paper [Xu et al.] have not compared with either PPO-Langrangian or CPO. They compare with a different algorithm called PDO and unconstrained TRPO (page 8 of [Xu et al.]). Furthermore, CRPO compares on the Cartpole and Acrobot environments, which are different benchmarks than those presented in our paper. Our implementation of CRPO baseline is based on that found in open-source python library omnisafe. Note that CRPO still inherits the limitations of the CMDP framework, namely it cannot operate in a state-wise hard constraint framework and the cost threshold needs fine-tuning and/or prior knowledge of the environment is needed.
>
> >Could you analyze the difference between RESPO and CRPO regarding the reward update and cost update?
>
> The optimization procedures in RESPO and CRPO use the reward and cost critics differently, though the algorithms' critic updates are similar. Particularly, our contributions are in our novel proposed reachability-based optimization framework which maintains a reward critic, cost critic, and reachability estimation function (REF). Our work addresses issues about guaranteeing as best as possible to maintain hard constraints in a stochastic environment and uses a primal-dual constraint optimization method. On the other hand, CRPO’s contribution is situated within a different optimization framework, namely Constrained Markov Decision Processes (which is not suitable for hard constraints, see CMDP section 3.2 in main paper). They propose a policy optimization technique that is primal based.
>
> >If the agent does not find the safe action, will it be stuck at a point?
>
> No, unless staying stationary is an available safe action. Based on our RESPO optimization, if there is no action so that the agent is currently or in the future safe, that means the current state is outside the feasible set. So, our optimization will be producing a policy that chooses trajectories minimizing the cumulative discounted sum of costs starting from that state. For a visual of an experiment showing how our algorithm might handle this, we refer the reviewer to the Double Integrator example in Appendix Section 4.4 in which the agent starts outside the feasible set.
>
> > How do you define the reachable set when considering reward performance?
>
> In definition 5 (in Section 4.1 of the main paper), we define the feasible set as the set of states from which no violation is reached under a given policy (the reachable violation set will be the complement of the feasible set). RESPO will optimize reward performance under the constraint that the cumulative cost is 0 (i.e. the agent remains in the feasible set) if the agent is in the feasible set. Furthermore it will minimize cumulative cost if the agent is not in the feasible set.
>
> >The balance between reward and cost is not addressed well, in the experiments, as shown in Figure 3, although RESPO can ensure safety, the trajectory is longer than other baselines, and also not smooth.
>
> In lines 318-322 of main paper, we describe how Figure 3 demonstrates RESPO has a desirable balance between reward and cost: it ensures the satisfaction of the various constraints as well as reaches the goal locations closer than the other approaches, therefore maintaining high rewards.
>
> Note that RESPO’s trajectory is actually the shortest ($\sim 75$ steps) compared to the other baselines ($\sim 85$, $80$, and $\sim 120$) as indicated in the bar “Trajectory Step Number” next to each plot in Figure 3.
>
> Additionally, the RESPO trajectory is quite smooth except for a particular point in the upper drone agent, which we explain in lines 320-322 of the main paper is to satisfy the hard constraints. The upper drone provides room to permit only one drone at a time in the tunnel, which is the intended behavior. The other approaches do not create this intended behavior of one drone at a time through the tunnel.
>
> >Some related papers are not mentioned in the study, e.g., [Kochdumper, N., et al., 2023], [Gu, S., et al., 2022] and [Selim, M., et al., 2022].
>
> Thank you, we will add these to our related works discussion. All these three papers rely on having access to, or recreating, the model of the system dynamics to make predictions about future environment rewards and/or response, and are therefore model-based while our approach falls within the category of model-free safe RL.
>
> [Xu et al.] Crpo: A new approach for safe reinforcement learning with convergence guarantee. In ICML. PMLR, 2021.

---

### Author Rebuttal · Authors · 2023-08-07

We thank the reviewers for their detailed suggestions and feedback. Our novel algorithmic contributions in this paper include 1) providing a reachability-based hard constraint satisfaction approach for stochastic and deterministic settings and 2) the ability of (re)entrance into the feasible set within our reachability-based framework. We have evaluated our proposed approach on several benchmarks and compared with various baselines. We also demonstrated our method can handle and prioritize multiple hard and soft constraints. We have provided a proof that our algorithm converges to the local optimum of our proposed optimization.

We have run additional experiments for a model-free learned CBF baseline and compared with our proposed approach and other baselines. We include figures of the results in the attached pdf.

---

### Decision · Program_Chairs · 2023-09-21

**Decision:**

Accept (poster)

**Comment:**

The meta-reviewer has reviewed the paper, the reviews, the responses, and agrees with the majority of the reviewers that this paper meets the NeurIPS standard.